# Trim66's paternal deficiency causes intrauterine overgrowth

Monika Mielnicka[1], Francesco Tabaro[1], Rahul Sureka[1], Basilia Acurzio[1], Renata Paoletti[2], Ferdinando Scavizzi[3], Marcello Raspa[3], Alvaro H Crevenna[1], Karine Lapouge[4], Kim Remans[4], Matthieu Boulard[1]

The tripartite motif-containing protein 66 (TRIM66, also known as TIF1-delta) is a PHD-Bromo–containing protein primarily expressed in post-meiotic male germ cells known as spermatids. Biophysical assays showed that the TRIM66 PHD-Bromodomain binds to H3 N-terminus only when lysine 4 is unmethylated. We addressed TRIM66's role in reproduction by loss-of-function genetics in the mouse. Males homozygous for *Trim66-null* mutations produced functional spermatozoa. Round spermatids lacking TRIM66 up-regulated a network of genes involved in histone acetylation and H3K4 methylation. Profiling of H3K4me3 patterns in the sperm produced by the *Trim66*-null mutant showed minor alterations below statistical significance. Unexpectedly, *Trim66*-null males, but not females, sired pups overweight at birth, hence revealing that *Trim66* mutations cause a paternal effect phenotype.

## Introduction

Sexual reproduction entails the mixing of the two parental genomes at each generation. In mammals, the male gametes are required to travel outside the body and survive for several days in the female oviduct to fertilize the oocyte. Through this journey, the paternal genome is embedded in an extremely compact form of chromatin. The packaging of the DNA in a volume less than 5% of a typical somatic cell nucleus enables a substantial reduction in the volume of the sperm head, hence enhancing sperm motility and penetration to the zona pellucida surrounding the egg (Chang et al, 2023). The highly condensed sperm chromatin is the result of a genome-scale remodeling of the chromatin composition occurring during spermiogenesis. This process initiates in post-meiotic haploid germ cells known as round spermatids, whereby lysine residues of histone tails get hyperacetylated, neutralizing their positive charge, which in turn

weakens their affinity for negatively charged DNA. In addition, acetylated histone tails recruit Bromodomain-containing proteins (Marmorstein & Zhou, 2014). The human genome encodes over 40 Bromodomain-containing proteins; in elongated spermatids, the protein Bromodomain testis associated (BRDT) binds to acetylated histone H4 and promotes the replacement of the vast majority of histones by small arginine-rich and cysteine-rich proteins called transition proteins and protamines (Prms) (Gaucher et al, 2012). As a consequence, Prms are the main DNA-binding proteins in mature spermatozoa; however, small quantities of nucleosomes are retained on specific DNA sequences (about 10% in human and 1% in mouse) (Gatewood et al, 1987; Hammoud et al, 2009). The retention of nucleosomes in sperm was reported to contribute to some degree to early development (Lismer & Kimmins, 2023). However, the mechanism that controls histone retention in sperm is unknown and the molecular details of chromatin reconfiguration post-meiosis remain incompletely understood.

This study addresses the biochemical and biological function of the Bromodomain-containing protein known as the *tripartite motif-containing protein 66* (TRIM66, also known as TIF1-delta), which is seemingly only expressed in post-meiotic male germ cells (Khetchoumian et al, 2004). TRIM66 is evolutionary-conserved (84% identity between mouse and human), and is also known as TIF1-delta because of its affiliation with the TIF1 (transcription intermediary factor 1) family that is defined by the presence of PHD-Bromo adjacent domains at the C-terminus. The N-termini of TIF1 proteins contain the coiled-coil and Bbox domains, similar to all other tripartite motif-containing proteins (Fig 1A). The TIF1 family is composed of four members: TIF1-alpha (also known as TRIM24), TIF1-beta (TRIM28, KAP1), TIF1-gamma (TRIM33), and TIF1-delta (TRIM66) (Zeng et al, 2008). TRIM66, TRIM24, and TRIM28 harbor a PxVxL motif (where x is any amino acid) that recruits heterochromatin protein 1 (HP1), thus suggesting a role in gene silencing (Khetchoumian et al, 2004; Zuo et al, 2022). The best characterized member of the TIF1 family is TRIM28, which plays an essential role in the repression of

[1]Epigenetics and Neurobiology Unit, EMBL Rome, European Molecular Biology Laboratory, Monterotondo, Italy [2]Plaisant s.r.l., Rome, Italy [3]National Research Council (IBBC), CNR-Campus International Development (EMMA-INFRAFRONTIER-IMPC), Monterotondo, Italy [4]European Molecular Biology Laboratory, Protein Expression and Purification Core Facility, Heidelberg, Germany

Correspondence: matthieu.boulard@embl.it

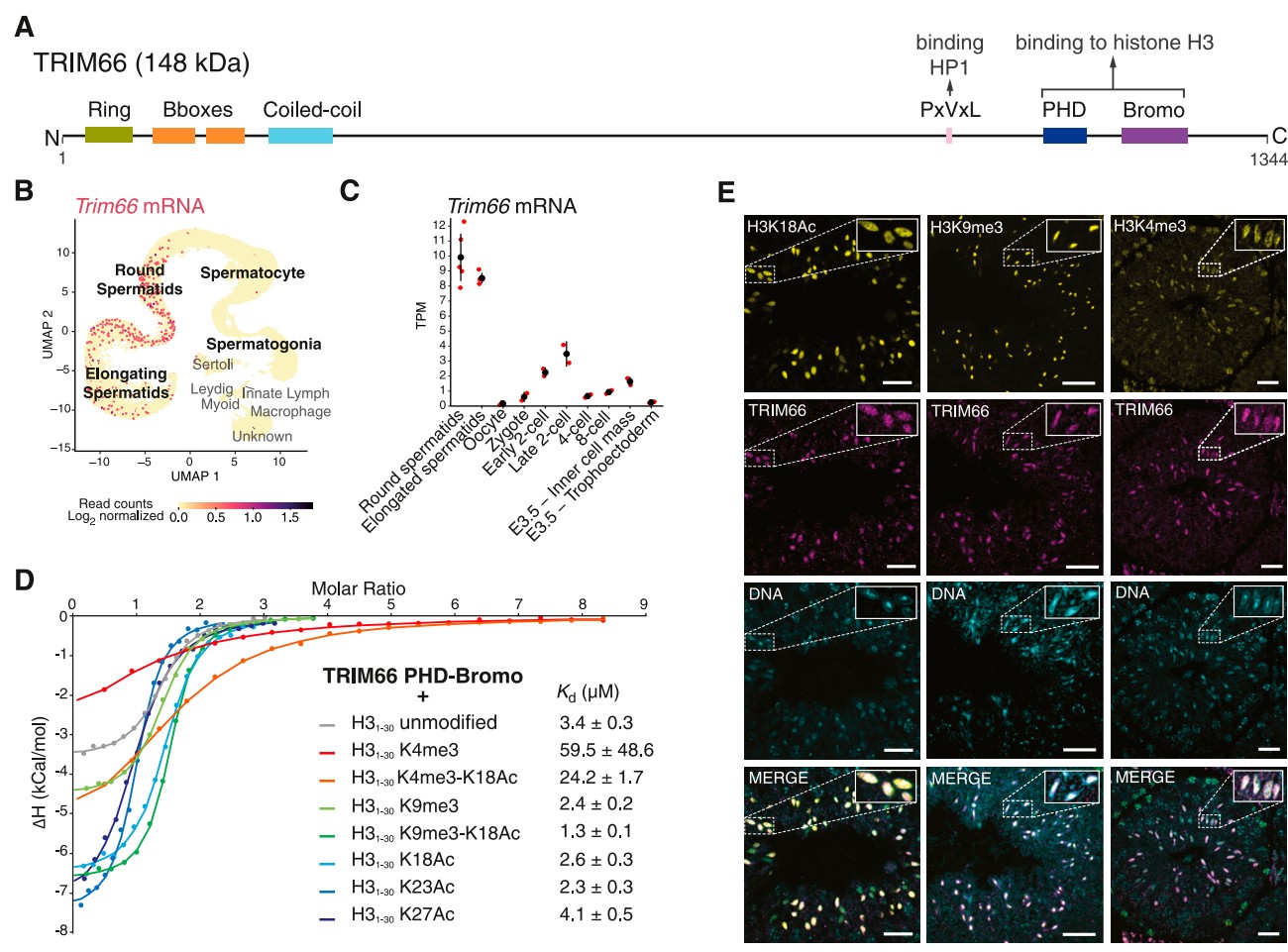

**Figure 1. TRIM66 is a spermatid-specific PHD-Bromo protein that recognizes unmethylated lysine 4 of histone H3.**
**(A)** Diagram of domain organization of murine TRIM66. The longest isoform containing a RING domain at the N-terminus is shown (NP_001164383.1). **(B)** *Trim66*'s expression in mouse adult testicular cells, as shown by Uniform Manifold Approximation and Projection of scRNA-seq data (GSE142585). The color scale represents the level of the expression of *Trim66* that is only detected at the post-meiotic stages, namely, round and elongated spermatids. **(C)** mRNA levels of *Trim66* in Transcripts Per Million in round and elongated spermatids and oocytes, and at key stages of preimplantation development. Embryo mRNA-seq data are from GSE66582 (Wu et al, 2016), and from this study for elongated and round spermatids. Individual biological replicates are shown in red and the average in black. The whiskers represent the mean Transcripts Per Million value plus or minus the SD. **(D)** Isothermal titration calorimetry measurements showed that TRIM66 PHD-Bromodomain binds to an unmodified histone H3 N-terminus tail ($K_d = 3.4\ \mu M$). A similar binding affinity is observed when lysine 9 is tri-methylated, when lysines 18, 23, or 27 are acetylated, or when lysine 9 is tri-methylated in combination with acetylated lysine 18. Tri-methylation of lysine 4 dramatically decreases the binding affinity of the TRIM66 PHD-Bromodomain to an H3 N-terminus tail. Combining the tri-methylated lysine 4 with acetylated lysine 18 only decreases the binding affinity by a factor of 7.1 when comparing to binding to an unmodified H3 histone tail, indicating a contribution of lysine 18 acetylation to the binding. **(E)** Representative confocal images of the indicated histone marks (yellow) and TRIM66 (purple) in the seminiferous epithelium. In elongated spermatids, TRIM66 is present both within and outside the chromocenter. The DNA is stained with Hoechst 33342. The post-translational modifications that influence TRIM66's binding to histone H3 (e.g., K4me3, K9me3, and K18Ac) show distinct patterns in elongated spermatids. The overlapping staining of TRIM66 and H3K18Ac and H3K9me3 is consistent with occupancy of H3K9me3, K18Ac nucleosomes by TRIM66. The scale bar is 25 µm.

retrotransposons and in the monoallelic expression of imprinted genes (Rowe et al, 2010; Alexander et al, 2015; Boulard et al, 2020). Structurally, the PHD-Bromodomain of TRIM28 differs from that of other TIF1s and acts as a small ubiquitin-like modifier (SUMO) E3 ligase (Zeng et al, 2008). Instead, the PHD-Bromodomains of both TRIM24 and TRIM33 recognize specific modifications of the N-terminus of histone H3: acetylation at K23 and K27 for TRIM24 and tri-methylation at K9 combined with acetylation at K18 for TRIM33. The binding of both TRIM24 and TRIM33 to H3 N-terminus is abolished by methylation at K4 (Tsai et al, 2010; Xi et al, 2011). The binding specificities of TRIM66

remain debated (Chen et al, 2019; Zuo et al, 2022). In this work, we provide evidence that the interaction of TRIM66 PHD-Bromo with H3 N-terminus requires K4 to be unmethylated, similarly to TRIM33 and TRIM24. To understand the biological role of TRIM66 in spermiogenesis and reproduction, we created two independent *Trim66* loss-of-function mutations in the mouse. Homozygous *Trim66*-null mutations did not measurably impact sperm mobility nor fertilization capacity. However, *Trim66*-mutant males sired pups overweight at birth. The data thus revealed that *Trim66*'s paternal deficiency causes viable intrauterine overgrowth.

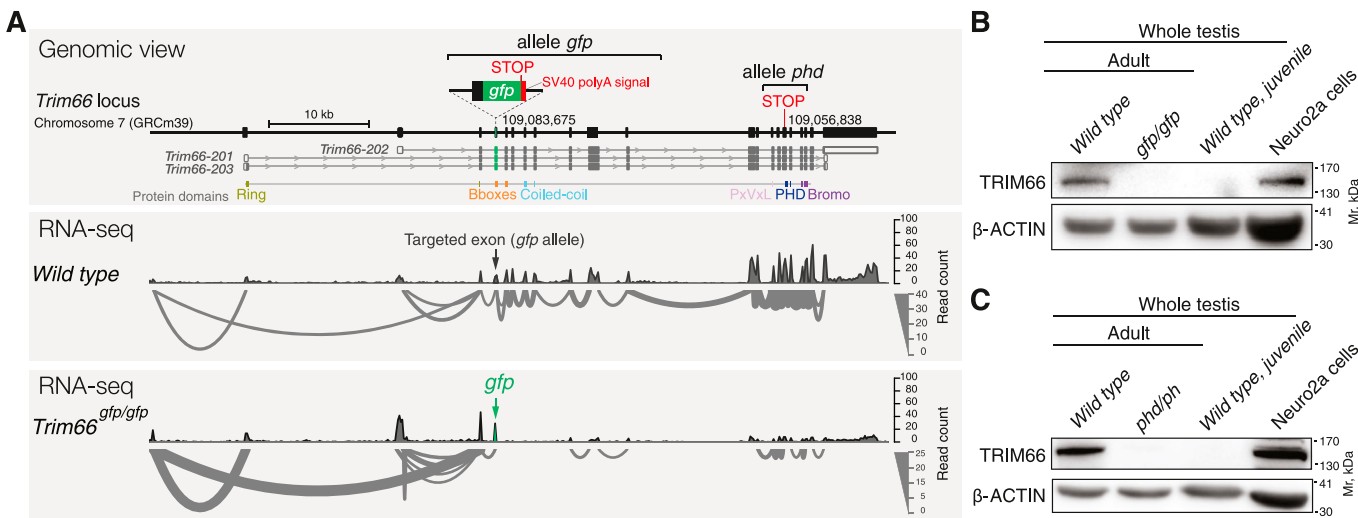

**Figure 2. Creation of two *Trim66*'s loss-of-function murine alleles.**
**(A)** Schematic representation of the genomic structure of the murine *Trim66* gene and location of the CRISPR-mediated insertions of the alleles *Trim66^gfp* and *Trim66^phd* (upper panel). The three coding mRNA isoforms are shown, as well as their translation (upper panel). The middle and bottom panels show splicing patterns as inferred from RNA-seq in sorted round spermatids. Two independent loss-of-function alleles were created: *Trim66^gfp* and *Trim66^phd*. *Trim66^gfp* consists of the insertion of a premature stop codon and poly-A signal in exon 3 (*Trim66-202*) resulting in truncated transcription as shown with the sashimi plot (lower panel). No alternative splicing event is detected in the homozygous mutant. The allele *Trim66^phd* was created by insertion of a premature stop codon in exon 15 (*Trim66-201*). The genomic coordinates of the insertions are indicated (GRCm39). **(B)** Western blot detection of TRIM66 in whole testis extracts and Neuro2A cell line (positive control) using polyclonal antibodies raised against murine TRIM66's Bromodomain. Detection of TRIM66 in adult but not in juvenile testes is consistent with its expression in post-meiotic germ cells. TRIM66 protein is undetectable in the lysate from adult testes homozygous for *Trim66^gfp*, thus supporting a complete loss of function. **(C)** Western blot with the antibody anti-TRIM66 showing that TRIM66 is undetectable in adult testes homozygous for *Trim66^phd*, arguing that the insertion of a stop codon in exon 15 is a bona fide null mutation. Source data are available for this figure.

# Results

## TRIM66 is a PHD-Bromo–containing protein with spermatid-specific expression

The seminal study that uncovered *Trim66* as a paralogous gene of *Trim28*, *Trim24*, and *Trim33* suggested that its expression could be restricted to the testis (Khetchoumian et al, 2004). A more extensive survey of *Trim66*'s expression in 33 mouse tissues using the 5′ RNA FANTOM5 dataset confirmed that the testis is the only assessed organ with high levels of *Trim66* mRNA (Lizio et al, 2015) (Fig S1A). Furthermore, analyses of murine and human testicular single-cell RNA-seq data show that *Trim66* mRNA is only transcribed in post-meiotic germ cells, namely, round and elongated spermatids in both species (Figs 1B and S1B). Noticeably, *Trim66* mRNA was undetectable in the ovary (Figs 1C and S1B). A recent study proposed that TRIM66 could act as a negative regulator of totipotency, but its expression in the early embryo has not been reported (Zuo et al, 2022). Thus, we computed *Trim66* mRNA levels at all key stages of preimplantation development using public RNA-seq data (Wu et al, 2016). After fertilization, *Trim66* starts to be transcribed at low levels at the two-cell stage, presumably during zygotic genome activation. At the four-cell stage, its expression drops to background levels and remains barely detectable at subsequent cleavage stages (Fig 1C). Our survey of bulk and single-cell RNA-seq data concluded that *Trim66* is primarily expressed in round and elongated spermatids and is undetectable in the oocyte.

Three murine *Trim66* mRNA isoforms produced by alternative promoters are documented (Fig 2A): The longest transcript (e.g., *Trim66-203*) encodes a RING finger domain at the N-terminus that is absent in the two other isoforms, because of their transcription from an internal promoter (e.g., *Trim66-201* and *Trim66-202*). All three known isoforms possess the defining domains of TIF1 proteins (e.g., tandem Bbox, coiled-coil, and PHD-Bromo). To gain further insights into the *Trim66* mRNA isoforms expressed in the testis, we performed 5′ RACE experiments. Although our analysis failed to detect *Trim66-203*, we identified a previously unreported isoform that contains an additional non-coding exon (Fig S1C). The data indicate that the *Trim66* isoforms expressed in spermatids lack the RING domain.

## TRIM66 PHD-Bromo interacts with a histone H3 tail that is unmethylated at lysine 4

Previous biophysical studies about the histone binding specificities of TRIM66's PHD-Bromodomain led to conflicting results: Chen et al reported a specific interaction with histone H3 unmodified at R2 and K4 and acetylated at K56 (Chen et al, 2019), whereas Zuo et al found a specific binding to unmodified H3K4, tri-methylated K9, and acetylated K18 (Zuo et al, 2022). A possible cause for this disagreement could be the use of chimeric histone peptides by Chen et al (2019). To resolve this discrepancy, we measured the binding affinity of a recombinant PHD-Bromodomain protein with 30-mer H3 peptides bearing key post-translational modifications (PTMs), singly or in combination (Fig 1D). Our isothermal titration calorimetry experiments measured binding of the TRIM66 PHD-Bromodomain to the unmodified H3 N-terminus peptide with a

$K_d$ = 3.4 $\mu M$. Tri-methylation of lysine 4 (H3K4me3) dramatically decreased the $K_d$ to 59.5 $\mu M$, thus disrupting the interaction. Tri-methylation at K9, and acetylation at K18, K23, and K27 had no substantial influence on the binding affinity. The combination of the acetylated K18 (K18Ac) and H3K4me3 increased the binding affinity by a factor of 2.4 compared with H3K4me3 alone, indicating some degree of contribution of K18 acetylation to the binding. The combination of K9me3 and K18Ac on the same peptide resulted in the highest binding affinity measured in our assays ($K_d$ = 1.3 $\mu M$, Fig S2A and B). Taken together, these biochemical studies identified the combination of PTMs of H3 that recruit TRIM66 PHD-Bromo as H3K4me0, K9me3, and K18ac. The modification that had the greatest impact is H3K4me3 that disrupts the interaction. These results are in perfect agreement with Zuo et al (2022).

The association between K9me3 and K18Ac on the same H3 peptide appeared paradoxical, as K9me3 is a hallmark of heterochromatin, whereas histone acetylation typically causes chromatin opening. To assess the relative subnuclear localization of TRIM66 and H3's relevant PTMs, we used immunostaining and confocal microscopy and focused on elongated spermatids that can be unambiguously identified by their characteristic nuclear morphology (Fig 1E). In good agreement with a previous report, we detected TRIM66 protein in elongated spermatids, whereas no TRIM66 staining was observed in round spermatids (Khetchoumian et al, 2004). TRIM66 localized both within and outside the chromocenter. H3K9me3 staining was concentrated in the chromocenter, whereas H3K18Ac was mostly, but not exclusively, localized around it. Some nuclear territories were costained with TRIM66 and H3K9me3 and H3K18Ac. There was no global exclusion between H3K4me3 and TRIM66, suggesting that the binding inhibition may occur only locally.

## Creation of two independent *Trim66* loss-of-function murine alleles

To address the biological role of TRIM66 in vivo, we created two independent loss-of-function murine alleles using CRISPR/Cas9 genome engineering in the zygote (Fig 2A). The mutation termed *Trim66^gfp^* (FVB background) disrupts *Trim66*'s function by insertion of a premature stop codon and a SV40 poly-A signal in the first common exon of the three reported alternative transcripts (exon 3 of transcript *Trim66-201*). *Trim66^gfp^* was designed to abolish *Trim66*'s function, and at the same time to report its expression with the in-frame *egfp* cassette. Although the mutant allele transcribed mRNA with the *egfp* sequence (Fig S3A), no GFP expression could be detected in adult testicular cells (Fig S3B). Hence, the *egfp*-mRNA transcribed from the *Trim66^gfp^* allele appeared not to be translated in vivo, possibly because of the lack of proper 3′ UTR in the *egfp* chimeric transcript (Fig 2A) (Braun et al, 1989). We therefore used the *Trim66^gfp^* allele as a loss-of-function mutation. The second allele created, namely, *Trim66^phd^* (C57BL/6J background), disrupts *Trim66*'s function by insertion of a premature stop codon in exon 15 (transcript *Trim66-201*). Analysis of exon–exon junctions showed that splicing was not affected by the *gfp* insertion (*gfp* allele, Fig 2A), nor by the insertion of a premature stop codon (*phd* allele, Fig S4).

Next, we tested whether homozygosity for these mutations impacted the expression of TRIM66 at the protein level using polyclonal antibodies raised against TRIM66's Bromodomain. Western blot analysis detected TRIM66 protein in WT adult testes but not in juvenile testes, in agreement with its expression in post-meiotic germ cells that start to appear around postnatal day 25 (Fig 2B and C) (Khetchoumian et al, 2004). Importantly, the TRIM66 protein was undetectable in adult testes isolated from homozygous animals for both mutations, indicating a complete loss of function. Thus, *Trim66^gfp^* and *Trim66^phd^* are both bona fide null mutations (Fig 2B and C). For both mutations, heterozygous and homozygous animals were viable, fertile, and of normal visible phenotype.

## *Trim66*-mutant males, but not females, sire overweight progeny

When breeding homozygous *Trim66^gfp/gfp^* males with WT females, we unexpectedly observed that the progenies were overweight at birth (Fig 3A). The average weight of the naturally born pups sired by homozygous *Trim66^gfp/gfp^* males was 1.40 ± 0.187*g*, whereas the weight of the pups from WT control crosses was 1.31 ± 0.176*g* (*P* = 6.11 × $10^{-6}$, two-sided *t* test). Thus, pups sired by homozygous *Trim66^gfp/gfp^* males were on average 6.8% heavier than WT at birth. In contrast, no difference in weight was recorded for pups born from homozygous *Trim66^gfp/gfp^* mothers bred with WT males, indicating that the overgrowth is a paternal effect phenotype (Fig 3B). The size of the litters produced by homozygous *Trim66^gfp/gfp^* males and females was normal (Fig 3C and D). To rule out the possibility that this unanticipated phenotype could be caused by transcriptional disturbance other than *Trim66* loss of function (e.g., chimeric proteins or RNA in Trim66^gfp/gfp^ germ cells), we weighed the pups sired by *Trim66^phd/phd^* males (this mutation is a simple insertion of a premature stop codon; Fig 2A). The same intergenerational phenotype was observed with *Trim66^phd/phd^* homozygous males when crossed with WT females: the average weight of the born pups from *Trim66^phd/phd^* fathers was 1.38*g* ± 0.192 as compared to 1.34*g* ± 0.205 from the WT fathers (*P* = 0.0192, two-sided *t* test, Fig 3E). The variable severity of the phenotype could be caused by the different genetic backgrounds of the two loss-of-function strains (FVB versus C57BL/6J). Males homozygous for the *phd* mutation also produced litters of normal size (Fig 3F). The paternal effect overweight phenotype observed at birth (Fig 3A and E) did not persist until the weaning (Fig S5A and B). The overweight phenotype at birth was also observed when homozygous *gfp/gfp* males were crossed with homozygous *gfp/gfp* females (*P* = 4.94 × $10^{-6}$, two-sided *t* test, Fig S5C). However, intercrossing *gfp/gfp* males with *gfp/gfp* females resulted in litters with fewer pups at birth in comparison with isogenic WT breeding (*P* = 0.00678, two-sided *t* test, Fig S5D). This result suggests that disruption of both paternal and zygotic *Trim66* could cause a partially penetrant embryonic lethal phenotype.

Altogether, the data revealed that TRIM66's function in the male germline influences the weight of the progeny at birth.

## *Trim66* is dispensable for the maturation of functional spermatozoa

Given the specific expression of *Trim66* in round and elongated spermatids, we next examined the germ cells produced by *Trim66*-

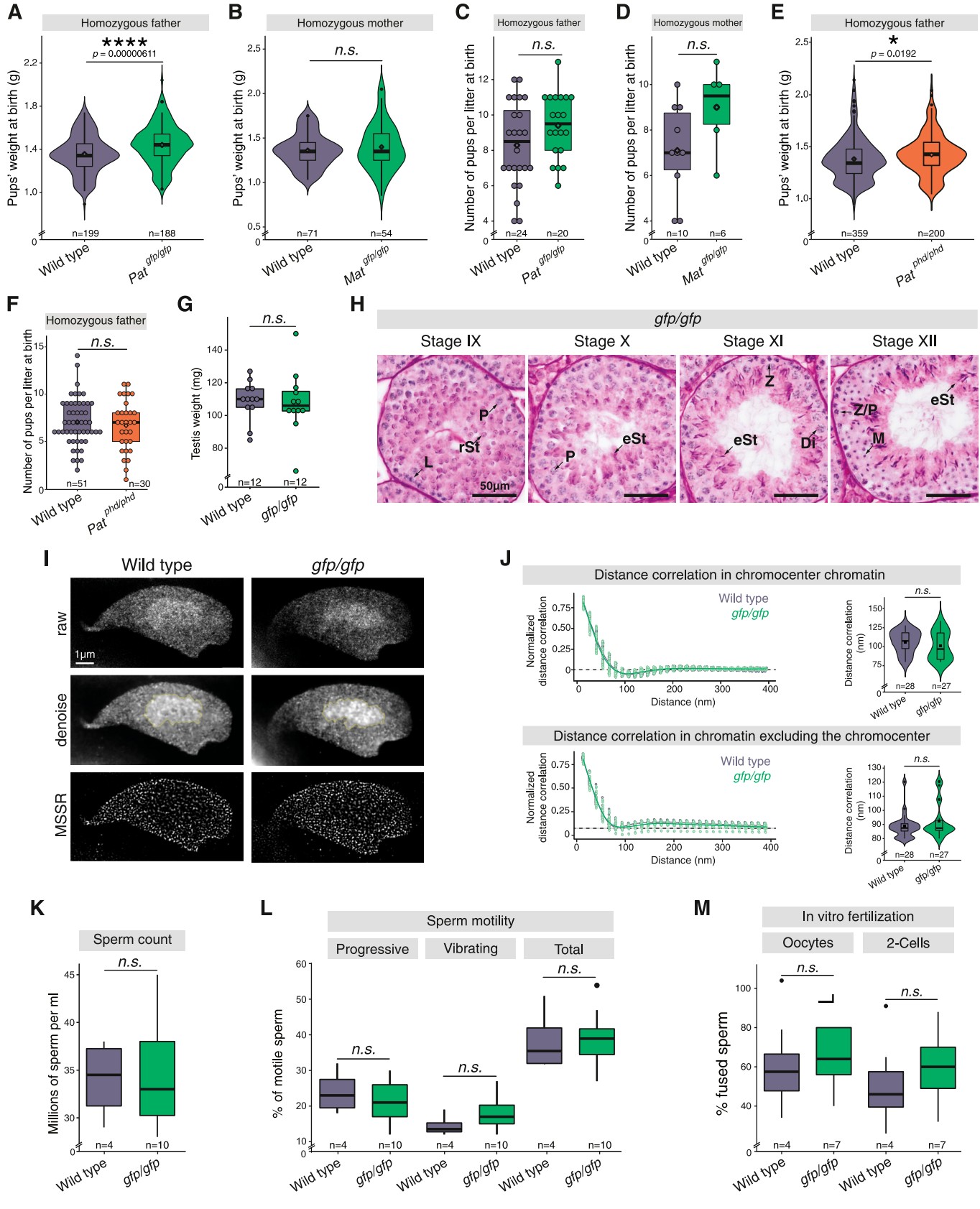

deficient males. Neither the testis weight nor the testis histology of *Trim66-mutant* males displayed any discernible abnormality (Fig 3G and H). The sperm heads produced by *gfp/gfp* males were morphologically normal (Fig S6A). We investigated higher order levels of chromatin organization in elongated spermatids by super-resolution imaging of spermatid DNA using stimulated emission depletion (STED) microscopy (Fig 3I). After DNA staining and super-resolution imaging, *Trim66*-deficient elongated spermatids were morphologically indistinguishable from the WT controls. We estimated the regularity of chromatin patterns by quantifying the signal distance correlation and found no statistical difference neither in the chromocenter nor in the rest of the nucleus (Fig 3J). Next, we performed functional assays on the sperm produced by *Trim66^{gfp/gfp}* homozygous mutants. Specifically, sperm count, motility, and capacity to fertilize oocytes by in vitro fertilization (IVF) showed no statistical differences between *Trim66^{gfp/gfp}* males and WT controls (Fig 3K–M). The same results were replicated for the *Trim66^{phd/phd}* (Fig S6B–E). In summary, the data show that the sperm produced by *Trim66*-deficient males display normal sperm concentration, motility, and fertilization efficiency.

## *Trim66*-deficient round spermatids up-regulate histone H3K4 methyltransferases

Previous reports have shown that TRIM66 acts as a transcriptional repressor in vitro (Khetchoumian et al, 2004; Zuo et al, 2022). Thus, we hypothesized that the viable overweight phenotype could originate from an abnormal mRNA payload in spermatozoa produced by *Trim66*-deficient males. We addressed this possibility by profiling total RNA levels in both round and elongated spermatids produced by *Trim66^{gfp/gfp}* homozygous males. Round and elongated spermatids were isolated from WT and *Trim66^{gfp/gfp}* animals by flow cytometry (Fig S7A–L), and their transcriptome was then analyzed by RNA sequencing of total RNA. Microscopic inspection of the sorted cells and clustering of the sorted cell populations based on the expression of marker genes confirmed the purity of the sorted round and elongated spermatids (Fig S7M–O).

We first examined the expression of transposable elements because it was reported that TRIM66 represses specific families of retrotransposons in mouse embryonic stem cells (e.g., L1Md_A and L1Md_T, MERVL-int and MT2_Mm) (Zuo et al, 2022). However, we

---

**Figure 3. *Trim66*-deficient males sire overweight progeny and produce functional spermatozoa.**
**(A)** *Trim66^{gfp/gfp}* homozygous males sire progeny overweight at birth. The violin plots represent the weight distribution of pups at birth sired by *Trim66^{gfp/gfp}* homozygous males bred with WT females. The boundaries of the overlaid box plot show the data within the first and third quantile, whiskers indicate minimum and maximum quartiles, the horizontal bar in the box plot shows the median, the rhombus indicates the mean, and the loose points show the data outliers. The *P*-values were calculated using a two-sided *t* test. 188 pups born from 5 *Trim66^{gfp/gfp}* homozygous fathers and 199 born from 5 WT fathers were weighed at birth. Pups sired by *Trim66^{gfp/gfp}* homozygous males were on average 6.9% heavier than WT controls at birth. **(B)** *Trim66*'s loss of function in females does not impact the weight of their progeny. The violin plots show the weight distribution of pups at birth, mothered by *Trim66^{gfp/gfp}* homozygous females bred with WT males. The boundaries of the overlaid box plot show the data within the first and third quantile, whiskers indicate minimum and maximum quartiles, the horizontal bar in the box plot shows the median, the rhombus indicates the mean, and the loose points show the data outliers. The *P*-values were calculated using a two-sided *t* test. 54 newborns from 3 *Trim66^{gfp/gfp}* mothers and 71 newborns from WT mothers were analyzed at birth. **(C)** Normal litter size sired by *Trim66^{gfp/gfp}* homozygous males when bred with WT females. The total number of litters with recorded pups number from 5 homozygous fathers was 20 and from 5 WT fathers was 24. Each litter size is shown as the single dot on the plot. The box plot boundaries show the data within the first and third quantile, whiskers indicate minimum and maximum quartiles, the horizontal bar in the box plot shows the median, and the rhombus indicates the mean. The *P*-values were calculated using a two-sided *t* test. **(D)** Normal litter size sired by *Trim66^{gfp/gfp}* homozygous females when bred with WT males. The total number of litters with recorded pups from homozygous mothers was six (born from three mothers) and from WT mothers was 10 (born from five mothers). Each litter size is shown as the single dot on the plot. The boundaries of the box plot show the data within the first and third quantile, whiskers indicate minimum and maximum quartiles, the horizontal bar in the box plot shows the median, and the rhombus indicates the mean. The *P*-values were calculated using a two-sided *t* test. **(E)** *Trim66^{phd/phd}* homozygous males sire progeny overweight at birth. The violin plots represent the weight distribution of pups at birth sired by *Trim66^{phd/phd}* homozygous males bred with WT females. The boundaries of the overlaid box plot show the data within the first and third quantile, whiskers indicate minimum and maximum quartiles, the horizontal bar in the box plot shows the median, the rhombus indicates the mean, and the loose points show the data outliers. The *P*-values were calculated using a two-sided *t* test. Two hundred pups born from 11 *Trim66^{phd/phd}* homozygous fathers and 359 born from 13 WT fathers were weighed at birth. **(F)** Normal litter size sired by *Trim66^{phd/phd}* homozygous males when bred with WT females. The total number of litters with recorded pups from 11 homozygous fathers was 31 and from 13 WT fathers was 51. Each litter size is shown as the single dot on the plot. The boundaries of the box plot show the data within the first and third quantile, whiskers indicate minimum and maximum quartiles, the horizontal bar in the box plot shows the median, and the rhombus indicates the mean. The *P*-values were calculated using a two-sided *t* test. **(G)** Average testicular weights in mg of adult mice of WT and *gfp/gfp* genotypes (n = 6). *P*-values were calculated using a two-sided *t* test. The boundaries of the box plot show the data within the first and third quantile, whiskers indicate minimum and maximum quartiles, the horizontal bar in the box plot shows the median, and the rhombus indicates the mean. The *P*-values were calculated using a two-sided *t* test. Each dot represents the weight of one testis collected from the six animals. **(H)** Histological sections stained with periodic acid–Schiff stain and hematoxylin of paraffin-embedded testes dissected from *Trim66^{gfp/gfp}* animals. The representative cell type for each stage is labeled with an arrow accordingly: rSt, round spermatid; eSt, elongating spermatid; P, pachytene spermatocyte; Di, diplotene spermatocyte; L, leptotene spermatocyte; Z, zygotene spermatocyte; Z/P, zygotene/pachytene spermatocyte; M, meiosis I and II cells. The scale bar is 50 μm. **(I)** Super-resolution imaging of high-order chromatin organization of elongated spermatids after DNA staining by stimulated emission depletion microscopy. Representative raw, denoised, and MSSR processed images are shown. The chromocenter is delimited with a yellow line in the denoise micrographs. WT and *Trim66^{gfp/gfp}* elongated spermatids are shown. The scale bar is 1 μm. **(J)** Quantification of the chromatin arrangements imaged by stimulated emission depletion microscopy. The calculated radial-averaged distance autocorrelation function for the DNA folding pattern in the chromocenter area (bottom right panel) and spermatid nucleus with exclusion of the chromocenter (bottom left panel). A regression line was calculated using a gamma function. The horizontal bar in the box in the box plot shows the median, and the square shows the mean. The *P*-values were calculated using a one-sided Wilcoxon test. The violin plot represents the data distribution for all collected points. **(K)** Caudal sperm count of 12-wk-old males for WT (n = 4) and *gfp/gfp* (n = 10). *P*-values were calculated using a two-sided *t* test. The boundaries of the box in the box plot show the data within the first and third quantile, the whiskers show the maximum and minimum quartiles, and the horizontal bar indicates the median. **(L)** Sperm mobility was assessed on animals aged 12 wk. The mature sperm was sourced from the cauda from WT (n = 4) and *gfp/gfp* (n = 7). *P*-values were calculated using a two-sided *t* test. The boundaries of the box in the box plot show the data within the first and third quantile, and the single dot shows data outliers. The whiskers show the maximum and minimum quantiles, and the horizontal bar indicates the median. **(M)** In vitro fertilization with sperm sourced from the cauda for WT (n = 4) and *gfp/gfp* (n = 7). The fertilized eggs were grown in vitro until they reached the two-cell stage. *P*-values were calculated using a two-sided *t* test. The boundaries of the box in the box plot show the data within the first and third quantile, and the single dot shows data outliers. The whiskers show the maximum and minimum quantiles, and the horizontal bar indicates the median.

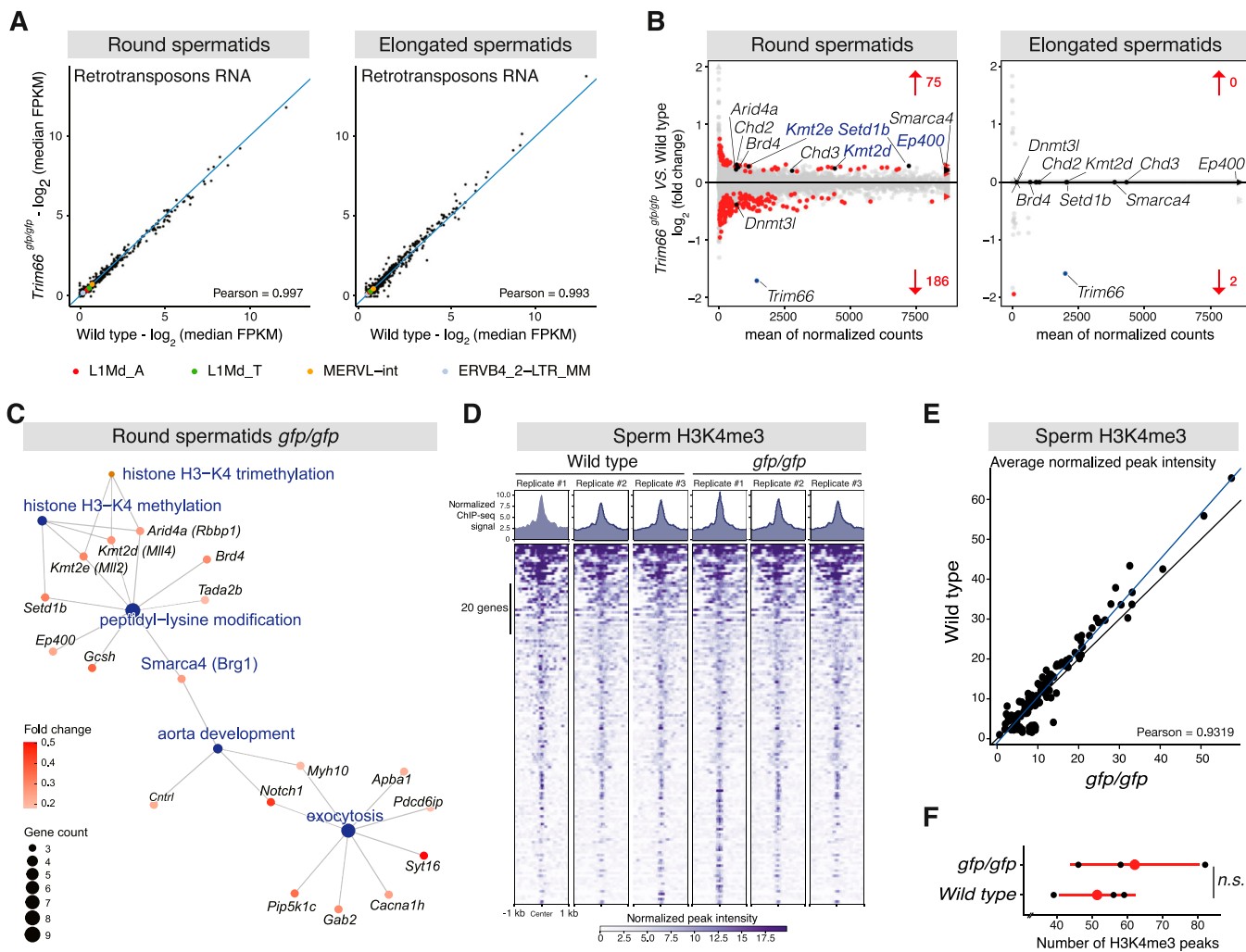

**Figure 4. Gene expression and H3K4 methylation changes in post-meiotic germ cells disrupted for *Trim66*.**
**(A)** Normal silencing of retrotransposons in *Trim66*-deficient round spermatids and elongated spermatids. Scatter plot comparing the expression of retrotransposons in WT and *Trim66*^gfp/gfp^ round spermatids (left panel) and elongated spermatids (right panel). No significant difference was observed. The family of retrotransposons previously reported to be reactivated in *Trim66* knockout mouse embryonic stem cells is shown (Zuo et al, 2022), and no change was detected in round spermatids and elongated spermatids lacking *Trim66*. **(B)** MA plots showing log₂ fold changes in gene expression (total RNA-seq) in *Trim66*^gfp/gfp^ round spermatids (left panel, n = 6) and *Trim66*^gfp/gfp^ elongated spermatids (right panel, n = 4). Significant gene expression changes are shown in red (*P* < 0.05, Wald's *t* test). Significantly up-regulated histone-modifying enzymes are shown: H3K4-specific histone methyltransferases *Set1b*, *Kmt2e*, *Kmt2d*, and the histone acetyltransferase *Ep400*. *Trim66* is significantly down-regulated and is labeled in blue. **(C)** Gene-concept network representing the results of the gene ontology functional enrichment analysis on significantly up-regulated genes. The graph shows the top five most enriched GO terms as blue nodes. The node size represents the number of significantly up-regulated genes annotated to a given term. Genes annotated to those terms are represented in shades of red. The color intensity represents the log₂ fold change. **(D)** Epigenomic profiling of H3K4me3 in spermatozoa isolated from WT and *Trim66*^gfp/gfp^ epidermis. Gene stack plots displaying ChIP-seq signal for all H3K4me3 peaks and average peak profiles in WT and *Trim66*^gfp/gfp^ sperm as indicated on the top. **(E)** Scatter plot comparing the average peak intensity of sperm H3K4me3 between *Trim66*^gfp/gfp^ and WT. Black line: true diagonal; blue line: linear regression fitted on the data. **(F)** Dot plot representing the number of H3K4me3 peaks in spermatozoa for the six analyzed samples. We detected on average 62 H3K4me3 peaks in *Trim66*-mutant sperm and 58 in WT. The slight increase in peak number in sperm chromatin of *Trim66*-mutant animal was not statistically significant (*P* = 0.44, *t* test).

found no evidence of retrotransposon activity in post-meiotic male germ cells disrupted for *Trim66* (Fig 4A).

We next assessed the impact of *Trim66*'s disruption on the expression of cellular single-copy genes and found that *Trim66* mutation impacted the expression of a relatively small number of genes in round spermatids: 186 genes were down-regulated and 75 up-regulated (adjusted *P* < 0.05; Fig 4B, left panel, Table S1). It is noteworthy that most of the deregulated genes are down-regulated, arguing that TRIM66 plays a more complex role than

repression in regulating gene expression in vivo. As expected, elongated spermatids displayed a greatly dampened transcriptional activity (Ernst et al, 2019), and *Trim66* itself was the only substantially deregulated gene (Fig 4B, right panel, Table S1). The absence of differentially expressed genes in elongated spermatids ruled out our initial hypothesis that *Trim66* disruption could lead to an ectopic accumulation of mRNA in spermatozoa. However, gene ontology analysis in round spermatids revealed that the function of up-regulated genes revolved around histone acetylation and H3K4

methylation (Fig 4C). The up-regulated gene set in *Trim66*-deficient round spermatids involved in histone acetylation included the histone acetyltransferase *Ep400*, the cofactor *NuA4*, and the Bromodomain-containing proteins *Brd4* (Fig S8). Up-regulated genes also included three histone methyltransferases that share their substrate specificity toward H3K4, namely, *Setd1b*, *Kmt2d*, and *Kmt2e* (Fig S8). This result raised the possibility that the coherent up-regulation of three H3K4-specific methyltransferases could alter H3K4 methylation patterns in round spermatids that may eventually propagate in the mature sperm if retained histones are affected.

### Minor H3K4me3 changes in the sperm of *Trim66*-deficient males

The hypothesis of sperm H3K4 methylation acting as an intergenerational information carrier has been proposed by previous studies showing that perturbations of sperm H3K4me3 result in developmental defects in subsequent generation(s) (Siklenka et al, 2015; Lismer et al, 2021a). Thus, we set out to profile H3K4me3 by ChIP-seq in the sperm produced by $Trim66^{gfp/gfp}$ and WT control males (Fig 4D). The average normalized H3K4me3 peak intensity was highly correlated between the WT and *Trim66*-null sperm in three biological replicates (Pearson's = 0.93, Fig 4E). Sperm produced by $Trim66^{gfp/gfp}$ males displayed a small number of ectopic peaks; however, statistical testing showed no significant difference ($P$ = 0.44, $t$ test, Fig 4F). Thus, the pattern of H3K4me3 in spermatozoa is not measurably impacted by loss of function of *Trim66*. Therefore, the paternally inherited signal that causes intrauterine overgrowth is likely other than sperm-retained H3K4me3, although it cannot be excluded that slight changes at key regulatory regions could influence early development.

## Discussion

This work provides evidence that loss-of-function mutations of the *Trim66* gene in the father cause viable intrauterine overgrowth of the progeny. This paternal effect phenotype was observed with high statistical significance with two independent mutations of *Trim66* on distinct genetic backgrounds, thus excluding strain-specific effects. Furthermore, this phenotype was reproducibly observed when homozygous males and females were intercrossed. The overweight phenotype of newborns was only observed for the progeny of homozygous fathers but not mothers, in good agreement with the specific expression of *Trim66* in round and elongated spermatids.

The paternal influence of *Trim66* on intrauterine growth observed in the mouse could be relevant for human health, given the high evolutionary conservation of *Trim66*. Furthermore, genome-wide association studies (GWAS) have identified *TRIM66* as a potential gene influencing the metabolism and obesity (NHGRI-EBI GWAS database, Table S2). The most frequently reported association is the influence of *TRIM66* on body mass index (22 studies of 44 where *TRIM66* was associated with a particular trait in GWAS) (Buniello et al, 2019).

Paternal effect phenotypes in mammals are rare and can have several possible causes (Rando, 2012; Lismer & Kimmins, 2023), for example, the loss of function of an imprinted gene normally expressed from the paternally inherited allele (Li et al, 1999). Recent transcriptome-wide assessments of imprinted gene expression using F1 hybrid mouse embryos, including in extraembryonic lineages, have not identified *Trim66* as monoallelically expressed (Andergassen et al, 2021; Edwards et al, 2023). It is therefore very unlikely that *Trim66* could be imprinted itself in the embryo. The paternally imprinted gene *H19* was previously shown to act on embryonic growth, but failure to methylate the *H19* imprinting control region in male germ cells was shown to cause growth retardation; thus, this possibility can be ruled out (Edwards et al, 2019).

In rare instances, paternal effect phenotypes have been reported in environmental perturbations and for a handful of genetic mutations that alter the epigenome of the male's gametes (Chong et al, 2007; Rando, 2012; Panzeri & Pospisilik, 2018; Lismer & Kimmins, 2023). Depending on the perturbation, two main mechanisms of genetic-independent paternal inheritance have been identified in the mouse: firstly, the transmission of small RNAs in the sperm payload, and secondly, altered H3K4me3 on the retained nucleosomes in the sperm. In addition, cytosine methylation has long been suggested to mediate intergenerational phenotypes; however, meticulous studies of robust paradigms found no evidence that supports this idea (Kazachenka et al, 2018; Galan et al, 2021).

Understanding the molecular mechanism of the overgrowth phenotype of pups sired by *Trim66*-null males is difficult because the molecular function of TRIM66 is scarcely understood. Insights can be gained from its biochemical properties. Of particular interest is the specific recognition of the N-terminus of histone H3 that is inhibited by methylation at K4. It is noteworthy that the binding of the PHD-Bromodomain of TRIM24, TRIM33, and TRIM66 to the H3 N-terminus is disrupted by K4 methylation; thus, the recognition of unmethylated H3K4 appears to be a unifying feature of TIF1 proteins, with the noticeable exception of TRIM28 (Zeng et al, 2008; Tsai et al, 2010; Xi et al, 2011; Zuo et al, 2022). Other chromatin-bound proteins were previously shown to recognize unmethylated H3K4. For instance, DNMT3L is an enzymatically inactive cofactor of de novo DNA methyltransferases, which binding to H3 is inhibited–methylated at K4; as a result, loci that are occupied by H3K4-methylated nucleosomes in prospermatogonia remain unmethylated, whereas the rest of the genome is densely de novo–methylated (Ooi et al, 2007). In the case of TRIM66, the biological consequence of the inhibition of its binding to H3 by K4 methylation remains unclear. Future profiling of TRIM66's genomic occupancy in spermatids should aid in clarifying this point.

The coherent up-regulation of H3K4-specific histone methyltransferase in round spermatids lacking functional *Trim66* raised the possibility that H3K4 methylation on the sperm-retained nucleosome could be the epigenetic defect passed to the next generation by *Trim66*-mutant males. However, the profiling of H3K4me3 in spermatozoa found no statistical differences between *Trim66* mutant and WT. This result does not completely exclude the possibility of H3K4me3 alteration in the sperm because antibody-based approaches for epigenomic profiling of the densely

condensed sperm genome are notorious to be imprecise (Yin et al, 2023).

The nature of the molecule transmitted from the father to the zygote causing the overgrowth phenotype remains to be identified. Potential candidates include small RNA species that were not surveyed in this study. On the contrary, it is noteworthy that a set of genes involved in exocytosis were up-regulated in round spermatids produced by Trim66$^{gfp/gfp}$ males (Fig 4C). Hence, an alternative possibility to histone modifications for the genetic-independent paternal influence on embryonic growth could be the transmission of extracellular vesicles in the seminal fluid.

# Materials and Methods

## Animal care and handling

All mouse procedures were done in accordance with EU Directive 2010/63/EU and under the approval of the Italian Ministry of Health (License 985/2020-PR to MB). Mice were housed in the pathogen-free Animal Care Facility at EMBL Rome on a 12-h light–dark cycle in temperature- and humidity-controlled conditions with free access to food and water.

## Generation of murine alleles

The mutations in the murine Trim66 gene were created by CRISPR/Cas9 editing technology as previously described (Quadros et al, 2017). For Trim66$^{gfp}$ (FVB/NCrl-Trim66$^{em2(gfp)}$Emr), a CRISPR crRNA oligo (TCTGCACATACTGCAACCGC) was annealed with tracrRNA and combined with a homology-flanked lssDNA donor containing an in-frame eGfp, stop codon, and SV40 poly-A signal. The target location was exon 3 transcript Trim66-201 (Ensembl v91); genomic coordinate: Chr 6. 109,083,675 (GRCm39/mm39). The insertion of a premature stop codon in the frame of the exon encoding the Bbox is expected to create a loss-of-function mutation (Table S3).

For Trim66$^{phd}$ (C57BL/6J-Trim66$^{em2(phd)}$Emr), a CRISPR crRNA oligo (GTGCGGTGTGCATCAACGGT) was annealed with tracrRNA and combined with a homology-flanked lssDNA donor containing a stop codon in-frame. The target location was exon 15 transcript Trim66-201 (Ensembl v91); genomic coordinate: Chr 6. 109,083,675 (GRCm39/mm39). The same protocol was followed for the two targeted alleles; briefly, the annealed sgRNAs were complexed with Cas9 protein and combined with their respective lssDNA donors (Cas9 protein 20 ng/ml, sgRNA 20 ng/ml, lssDNA 10 ng/ml, all IDT). These reagents were microinjected into zygote pronuclei using standard protocols (Du et al, 2019). Trim66$^{gfp}$ was produced using FVB zygotes, whereas Trim66$^{phd}$ was targeted in C57BL/6NJ zygotes (Charles River). After overnight culture, two-cell embryos were surgically implanted into the oviduct of day 0.5 post-coitum pseudopregnant CD1 mice. Founder mice were screened for by PCR, first using primers flanking the sgRNA cut sites, which identify InDels generated by NHEJ repair, and can also detect larger products implying HDR. Secondary 5′ and 3′ prime PCRs using the same primers in combination with template-specific primers allowed for the identification of potential founders; these PCR products were then Sanger-sequenced and aligned with the in silico design. Transgenic mouse production was performed by the Gene Editing and Virus Facility at EMBL Rome. Sequences of the single-stranded donor templates are provided in Table S3. All strains were maintained by breeding heterozygote animals with WT of the same genetic background. The Trim66$^{gfp}$ strain was maintained on a FVB/NCrl genetic background, and the Trim66$^{phd}$ strain was maintained on a C57BL/6J background.

## Genotyping

Genomic DNA was extracted from the tail biopsy using the PCRBIO Rapid Extract lysis kit (PCR Biosystems), and 4 µl of 1:2 dilution was used for PCR using 2X YourTaq TM Direct-Load PCR Master Mix (Biotechrabbit). The PCR conditions were as follows: initial denaturation step at 95°C for 2 min, then for the 30 cycles up to a cycle of incubation at 95°C for 10 s, 54°C for 10s, 72°C for 20 s, with a final extension step at 72°C for 5 min. Genotyping primers are listed in Table S3.

## Measurement of the weight of the progeny

Each Trim66$^{gfp}$ and Trim66$^{phd}$ homozygous male was paired at the age of 6 wk with a single WT female (age-matched between 6 and 8 wk at the start of the breeding assay). Conversely, Trim66$^{gfp}$ homozygous females were singly paired with a WT male. In parallel, the breeding of WT littermates was set up with WT animals of the corresponding genetic background. The females were exchanged for a younger one when they reached the age of around 30 wk. The cages were monitored daily for the new births, and the pups were weighed on the day of the birth in the morning. The pups were weighed again at the age of 20 d during weaning. The technician was unaware of the animals' genotype during the weight measurements. For the Trim66$^{gfp}$ males, up to five consecutive litters, and for males from Trim66$^{phd}$, up to four consecutive litters were tested statistically. For female Trim66$^{gfp}$, the data were collected for two consecutive litters.

## 5′ RACE analysis

The rapid amplification of 5′ cDNA end was carried out with the template switching reverse transcription protocol (Wulf et al, 2019) provided by New England Biolabs (NEB #M0466). In the first step, template switching RT reaction generated cDNAs with a universal template switching oligo (TSO), attached to the 3′ end of the cDNA. Reactions were set up according to the manufacturer's protocol. cDNAs served as a template for PCR amplification. Several PCRs were carried out with a common forward primer (TSO_PCR, annealing to the TSO) and a reverse primer specific to the gene of interest (Trim66, in exons 2 and 3). The PCR program was as follows: initial denaturation at 98°C for 30 s; amplification step 1 (5 cycles): 98°C for 10 s and 72°C for 30 s; amplification step 2 (5 cycles): 98°C for 10 s and 70°C for 30 s; amplification step 3 (30 cycles): 98°C for 10 s, 62°C for 15 s, and 72°C for 30 s; and final extension: 72°C for 5 min. The amplified pools were subjected to PCR product purification by Monarch PCR and DNA Cleanup Kit (NEB). The purified cDNAs were

cloned by NEB PCR Cloning Kit, and individual clones were isolated and sequenced (Azenta Life Sciences).

## Collection of sperm and sperm analysis

12-wk-old mice were euthanized by cervical dislocation, and each pair of epididymides was removed. One excised, epididymis was used for sperm analysis, and the other was used for in vitro fertilization (IVF). For sperm analysis, spermatozoa were released into 500 $\mu$l of preincubated human tubal fluid medium (HTF; Millipore, Merck KGaA) for 10 min at 37°C. Motility and concentration of sperm samples were determined using a Makler counting chamber (Sefi Medical Instruments) according to the World Health Organization guidelines (Makler, 1980; World Health Organization, 2010). A drop of 10 $\mu$l was placed on the chamber, and the number of spermatozoa was assessed and recorded using a microscope with a 20X objective.

The spermatozoa were scored in three groups: (1) progressive (spermatozoa exhibiting unidirectional movement), (2) vibrating (spermatozoa that show a strong movement in a stationary position), and (3) non-motile. The concentration was expressed as millions of spermatozoa per mL. Each time, at least six fields were counted with a minimum of 200 spermatozoa. All males were analyzed individually; six biological replicates were assessed.

## IVF assay

IVF was performed using the protocol previously described by Takeo and Nakagata (Takeo & Nakagata, 2011) and modified by Li et al (2016). For each IVF session, one epididymis from 12-wk-old WT and one from *Trim66-null* were transferred into 90 $\mu$l of capacitation medium, consisting of TYH (Takeo & Nakagata, 2011) with 0.75 mM methyl-$\beta$-cyclodextrin (MBCD; Sigma-Aldrich, Merck KGaA). Spermatozoa were allowed to disperse from the tissue and incubated for 30 min in a 5% $CO_2$ incubator at 37°C.

4-wk-old FVB or B6N females were previously superovulated by an intraperitoneal injection of 5 IU PMSG (Intervet) followed by 5 IU hCG (Intervet) 48 h later. At 12–14 h post-hCG injection, females were euthanized by cervical dislocation, their oviducts were removed, and cumulus–oocyte complexes (COCs) were incubated for 20 min into a fertilization drop, consisting of 250 $\mu$l of HTF and 1 mM reduced L-glutathione (GSH; Sigma-Aldrich, Merck KGaA). To reduce the female-to-female variability, the COCs from each female were divided between the two experimental groups.

After capacitation, 8 $\mu$l of spermatozoa was collected from the peripheral part of each capacitation drop and transferred to inseminate the COCs (final sperm concentration 2–6 × $10^5$ spermatozoa/ml). After 4 h, the oocytes inseminated were washed three times in 200 $\mu$l of HTF medium and cultured overnight. 24 h after insemination, the IVF rate, expressed as the percentage of two-cell embryos obtained with respect to the number of total oocytes, was determined. IVF was performed with sperm isolated from six males homozygous for the GFP mutation and six WT controls, each using COCs from 18 females.

## FACS

From male *Trim66^{gfp}* and WT littermates aged 23–24 wk, round and elongated spermatids were isolated from the same testes by FACS using a BD FACSAria II (BD Biosciences) with slight modifications from a previously published protocol (Bastos et al, 2005). Briefly, the material was sourced from both testes for each mouse and was run until the whole material was sorted. The dissected tissue was incubated in enhanced Krebs–Ringer's buffer (120 mM NaCl, 4.8 mM KCl, 25.2 mM $NaHCO_3$, 1.2 mM $KH_2PO_3$, 1.2 mM $MgSO_4$, 1.3 mM $CaCl_2$, 11 mM glucose, 1X non-essential amino acids [Gibco], Pen/Strep 1X [Gibco]). De-encapsulated testes were digested for 5 min in 0.5 mg/ml collagenase (*Clostridium histolyticum*, Sigma-Aldrich), then treated for 10 min with trypsin (acetylated from bovine pancreas, type V-S) at 32°C. The staining was done in EKRB buffer with 10% FBS, with 10 $\mu$g/ml Hoechst 33342 for 30 min and with 2 $\mu$g/ml propidium iodide at 32°C for 10 min. The cells were sorted in a sorting buffer (10% FBS enhanced Krebs–Ringer's buffer). The sorted spermatid fractions were collected in Monarch DNA/RNA Protection Reagent (NEB) and flash-frozen in liquid nitrogen before RNA extraction.

## RNA extraction and library preparation

Total RNA was extracted from collected spermatid fractions with Monarch RNA Extraction Kit (NEB) according to the manufacturer's protocol for RNA extraction from tissue or leukocytes, with proteinase K incubation time optimized for homogenized tissues. The optional step with DNase I treatment was performed, with an additional treatment with Turbo DNase to eliminate all traces of genomic DNA (Invitrogen). Total RNA was then purified with Monarch RNA Cleanup Kit (NEB) and quantified using Qubit Fluorometric Quantification (Thermo Fisher Scientific). The integrity of the RNA was determined using the TapeStation High Sensitivity RNA kit (Agilent Technologies). 10 ng of total RNA has been used to prepare ribo-depleted RNA-seq libraries with these kits: NEBNext rRNA Depletion Kit (Human/Mouse/Rat) and NEBNext Ultra II Directional RNA Library Prep Kit for Illumina. We used 17 PCR cycles to obtain enough material for sequencing. Obtained NGS libraries were pooled in equimolar amounts and sequenced using the Illumina NextSeq 500 sequencer with a 40PE running mode.

## Histology

Testes were dissected on 55-wk-old animals and incubated for 24 h in Bouin's solution (Sigma-Aldrich) at 4°C. After fixation, the samples were dehydrated and embedded in paraffin blocks. The samples were sectioned to produce 10-$\mu$m-thick sections. The periodic acid–Schiff (PAS) staining method was adapted from Ahmed and de Rooij (2009) with slight modifications to stain for acrosome formation in testis sections (Ahmed & de Rooij, 2009). The oxidation step with 1% periodic acid was for 10 min. The Schiff reagent (Sigma-Aldrich) incubation was for 20 min. The Harris hematoxylin was added to stain the cells alongside the acrosome. The samples were imaged at 40X magnification with a color camera slide scanner microscope (Olympus).

## Western blot

Immunodetection of TRIM66 was performed on a whole testis lysate from adult males (35 and 34 wk old for WT and *gfp/gfp* homozygous mutants, respectively) and juvenile mice (15 d old). The de-encapsulated testes were mechanically grinded in RIPA buffer (150 mM NaCl, 1% NP-40, 0.5% deoxycholic acid, 0.1% SDS, 50 mM Tris, pH 8.0) until the complete dissolvement of the tissue inside the buffer. The protein sample was then diluted to the appropriate concentration in SDS loading buffer (50 mM Tris–HCl, pH 6.8, 2% SDS [Sigma-Aldrich], 10% glycerol [Sigma-Aldrich], 100 mM DTT, 0.1% bromophenol blue [Sigma-Aldrich]) and heated–denatured at 95°C for 5 min. The samples were electroporated in the MES buffer in 1 mm 4–12% Bis-Tris gels (Invitrogen) and blotted onto a PVDF membrane using Trans-Blot Turbo Transfer System (Bio-Rad) for 15 min. The membrane was stained with 0.1% (wt/vol) Ponceau S in 5% (vol/vol) acetic acid solution for 5 min, washed with PBS, and blocked with milk (5% dry milk powder [Roth], 0.1% Tween [Sigma-Aldrich]). Primary antibodies were incubated in blocking buffer overnight at 4°C. The concentration of TRIM66-Bromodomain antibody was 1:50, and *β*-actin, 1:7,500; the membrane was then washed five times for 5 min in TBS-T; secondary antibodies HRP-conjugated (Goat anti-Rabbit IgG [H+L] Secondary Antibody, 31460; Invitrogen) were incubated for 1 h at RT in blocking buffer, and the membrane was washed five times for 5 min in TBS-T. Detection was performed using ECL (GE Healthcare) according to the manufacturer's protocol, and images were acquired using an imager Amersham ImageQuant 800 (GE Healthcare). Regions of relevant molecular weight were cropped for presentation.

## Immunofluorescence

The freshly dissected testes from 23- to 25-wk-old males were fixed with modified Davidson's fixative overnight at 4°C. The testis tissues were then dehydrated in ethanol series: twice for 30 min in 50% and 70%, and stored in 70% ethanol before processing. The stored tissue samples in 70% EtOH were washed twice for 1 h in 96% and 100% ethanol at 4°C. The ethanol-washed samples were then incubated three times in xylene for 30 min, then washed twice with paraffin at 56°C, and then incubated with melted paraffin at 56°C overnight before transfer to the tissue mold. The samples were cooled for 1 d before making 7-*μm* sections using a microtome. The sections were dried at 42°C overnight before immunostaining.

The sections were dewaxed in xylene twice for 10 min and then rehydrated in the ethanol series of two washes in 100% EtOH for 5 min, and a single wash in 96% EtOH for 2 min, 70% EtOH for 2 min, and 50% EtOH for 2 min, followed by two washes in distilled water for 5 min before acid/heat/antigen retrieval step. The antigen retrieval was performed in the microwave (730 W) for 10 min in 10 mM citrate buffer, pH 6. Cooled slides were washed in PBS twice for 5 min, then permeabilized with 0.3% Triton X-100 in TBS for 10 min. The slides were washed three times for 5 min in 0.1% Triton X-100 in TBS. The blocking of the slides was done in 5% natural donkey serum in the TBS buffer with 0.1% Triton X-100 for 1 h at RT. The slides with antibodies were incubated overnight at 4°C in the blocking solution. The TRIM66 antibody (TIF1*δ*, PG124) (Khetchoumian et al, 2004) dilution was 1:250, and all histone

modification dilution was 1:200 (H3K4me3 [C15410003; Diagenode], H3K18Ac [ab1191; Abcam], H3K9me3 [ab8898; Abcam]). TRIM66 was detected with secondary donkey Alexa Fluor 647 (Thermo Fisher Scientific) anti-mouse antibody at 1:1,000 dilution, and all histone marks, with secondary donkey Alexa Fluor 546 (Thermo Fisher Scientific) anti-rabbit at 1:1,000 dilution. The slides were cured overnight with ProLong Glass Antifade Mountant with NucBlue Stain. The imagining was done with confocal microscopy at 60X magnification with an AX microscope (Nikon) with a galvano scanner. The final images were denoised to increase signal to background signal with the 4 Noise2Void plugin in Fiji (Krull et al, 2018 *Preprint*).

## Generation of polyclonal antibodies against murine TRIM66

Polyclonal antibodies were raised against a recombinant Bromo-domain of murine TRIM66 (amino acids 1,042–1,230) produced in *Escherichia coli*. Polyclonal antibodies against the purified TRIM66-Bromo were raised in a New Zealand White rabbit at the EMBL Laboratory Animal Resources Facility. After the immunization process, the rabbit was euthanized by exsanguination. The serum was isolated from the final bleed sample and used for antibody purification. The purified TRIM66-Bromo protein was covalently coupled to NHS-activated agarose beads (Pierce) according to the manufacturer's specifications. The serum was diluted 1:1 in PBS and incubated overnight with the TRIM66-Bromo resin pre-equilibrated in PBS. After overnight incubation, the resin was washed with PBS and the TRIM66-Bromo–specific antibodies were eluted with 100 mM glycine, pH 2.4, and 150 mM NaCl. The elution fractions were immediately neutralized with 1 M Tris, pH 8.5. The elution fractions were analyzed by SDS–PAGE, and the fractions containing antibodies were pooled.

## STED microscopy

The whole testes were frozen in OCT and sectioned to produce 12-*μm*-thick sections from 24-wk-old WT and *Trim66^{gfp/gfp}* animals. The sections were fixed with 4% PFA (Sigma-Aldrich) in PBS, stained with Hoechst/rhodamine dye for 10 min, then mounted in ProLong Diamond (Thermo Fisher Scientific), and left to cure overnight in the dark at 4°C. STED microscopy was performed using a STEDYCON (Abberior), using a 100x 1.45 NA oil immersion objective (Zeiss). The dwell time for both confocal and STED microscopy was 10 μs, whereas 15 line accumulations were used for STED. Several planes were imaged per cell using a step size of 0.25 *μm*. Denoising was performed using the Noise2Void plugin in Fiji (Krull et al, 2018 *Preprint*). Training was carried out for 300 epochs using all images from WT samples. The best network was saved and used to denoise all samples. Mean-Shift Super-Resolution (MSSR) was applied to selected denoised planes to increase resolution and remove background (García et al, 2021 *Preprint*). The Fiji plugin of MSSR was used to carry out the image processing. The radial-averaged autocorrelation was calculated using the available script at https://imagejdocu.list.lu/macro/radially_averaged_autocorrelation. The images then were processed using the Fuji 1.53f51 version threshold function to select for the nucleus and the chromocenter region inside the cell nucleus. The outside of the chromocenter area was

selected by subtracting the selected nucleus area from the selected chromocenter area of each image. Data were then exported and post-processed using custom scripts written in R. The data were modeled using the generalized additive model for the first 40 pixels. The points that were used for the statistical analysis of the radial-averaged autocorrelation were approximated to the first minimum of the modeled generalized additive model function. All points that were measured with the Fuji script for the radial-averaged auto-correlation and that were falling into previously described criteria were then plotted and tested with a two-sided Mann–Whitney test. For the chromocenter, the autocorrelation distance was estimated as the value where the gamma fitted regression line reaches 0 (bottom left panel). For the non-chromocenter chromatin, the autocorrelation distance was estimated as the value where the gamma fitted regression line reaches the minimum point.

## Dissection and sperm isolation

Epididymal sperm was isolated as previously described (Lismer et al, 2021b). Briefly, 39-wk-old mice were euthanized and the epididymis was dissected quickly. The epididymis was washed with ice-cold PBS, and the cauda was dissected from the rest of the epididymis. The caudae were then incubated in 1 ml of Donner's solution (25 mM NaHCO3, 1 mM sodium pyruvate, 0.53% sodium lactate, 2% BSA) at 37°C with gentle shaking for 1 h after making 3–5 incisions in the cauda to allow for the sperm to swim out. After they swam out, the sperm solution was filtered through a 40-$\mu$m nylon mesh, washed in PBS, and frozen in 200 $\mu$l of freezing media (Catalog# 90128; Irvine Scientific) at −80°C.

## Native ChIP-seq H3K4me3 in sperm and library preparation

Sperm ChIP-seq was performed as previously described (Lismer et al, 2021b). Briefly, sperm were thawed, washed with PBS, and counted using a hemocytometer. Eight million spermatozoa per sample were incubated with 20mM DTT at 20–25°C for 2 h. The DTT was quenched using 100 mM NEM (N-ethylmaleimide) and incubated for another 30 min at 20–25°C. To prepare chromatin, sperm were washed in PBS and resuspended in 200 $\mu$l complete buffer 1.1 (15.88 mM Tris–HCl, pH 7.5, 61.14 mM KCl, 5.26 mM MgCl$_2$, 011 mM EGTA, 0.5 mM DTT, 0.3 M sucrose). 200 $\mu$l of complete buffer 1.2 (15.88 mM Tris–HCl, pH 7.5, 61.14 mM KCl, 5.26 mM MgCl$_2$, 011 mM EGTA, 1.25% NP-40, 1% DOC) was added to the resuspended sperm, mixed by pipetting, and incubated on ice for 25 min. 400 $\mu$l MNase complete buffer (85 mM Tris–HCl, pH 7.5, 3 mM MgCl$_2$, 2 mM CaCl$_2$, 0.3 M sucrose, 60 IU of MNase [Catalog# 0247S; NEB]) was added to each chromatin preparation, mixed by pipetting, and incubated at 37°C for 5 min. MNase digestion was stopped by adding 2 $\mu$l of 0.5 M EDTA and mixing. At this point, the protease inhibitor (Catalog# 4693116001; Roche) was added to the digested chromatin to preserve chromatin-bound proteins. Protein G Dynabeads (Catalog# 10003D; Thermo Fisher Scientific) were washed, blocked with 0.5% BSA, resuspended in 50 $\mu$l combined buffer (50 mM Tris–HCl, pH 7.5, 30 mM KCl, 4 mM MgCl$_2$, 0.05 mM EGTA, 1 mM CaCl$_2$, 0.3 M sucrose), and added to digested chromatin solution for preclearing. The mixture was incubated at 4°C for 1.5 h on a rotator. The preclearing Dynabeads were removed by placing the tubes on a magnetic rack.

The supernatant was transferred to a fresh tube containing H3K4me3 antibody (Catalog# 15410003; Diagenode)–bound protein G Dynabeads that were washed and resuspended in 100 $\mu$l of combined buffer. The mixture was incubated at 4°C on a rotator for 14 h. The antibody-bound chromatin was washed with wash buffer A (50 mM Tris–HCl, pH 7.5, 10 mM EDTA, 75 mM NaCl), followed by a wash with wash buffer B (50 mM Tris–HCl, pH 7.5, 10 mM EDTA, 125 mM NaCl). The washed antibody–bound chromatin was eluted by heating the Dynabeads–antibody complex at 65°C with gentle agitation for 10 min in elution buffer (1X TE buffer, 5 mM DTT, 100 mM NaHCO$_3$, 0.2% SDS). The supernatant was transferred to DNA LoBind tubes, and digested with RNase and Proteinase K, and the DNA was cleaned and concentrated using ChIP DNA Clean and Concentrator Kit (Catalog# D5201; Zymo Research) according to the manufacturer's instructions. The sequencing libraries were prepared using NEBNext Ultra II DNA Library Prep Kit for Illumina (Catalog# E7645S; NEB) following the manufacturer's instructions.

## RNA-seq data analyses

Biological replicates are germ cells isolated from different animals. The quality of RNA-seq libraries was assessed with FastQC version 0.11.9 (https://www.bioinformatics.babraham.ac.uk/projects/fastqc/). Sequencing adapters were removed with Trimmomatic version 0.39 (Bolger et al, 2014) with the following parameters: ILLUMINACLIP:/opt/Trimmomatic-0.39/adapters/TruSeq3-PE.fa:1: 30:15:2:true SLIDINGWINDOW:20:22 MAXINFO:20:0.6 LEADING:22 TRAILING:20 MINLEN:40. Trimmed reads were aligned against the mouse genome (Gencode GRCm38 M25/mm10) using STAR version 2.7.6a (Dobin et al, 2013) and the following parameters: --seedSearchStartLmax 30 --outFilterMismatchNoverReadLmax 0.04 --winAnchorMultimapNmax 40. Duplicated reads were marked with Picard MarkDuplicates version 2.23.8.

The expression of single-copy genes and lncRNA was quantified using STAR's GeneCounts mode. The reference annotation was downloaded from the Mouse Genome Informatics resource. Differential expression was detected with DESeq2 version 1.30.1 (Love et al, 2014).

For transposable element expression, trimmed reads were aligned to the mouse genome (Gencode GRCm38 M25/mm10) with STAR version 2.7.6a and the following parameters: --outFilterMultimapNmax 5,000 --outSAMmultNmax 1 --outFilterMismatchNmax 3 --winAnchorMultimapNmax 5,000 --alignEndsType EndToEnd --alignIntronMax 1 --alignMatesGapMax 350 --seedSearchStartLmax 30 --alignTranscriptsPerReadNmax 30,000 --alignWindowsPerReadNmax 30,000 --alignTranscriptsPerWindowNmax 300 --seedPerReadNmax 3,000 --seedPerWindowNmax 300 --seedNoneLociPerWindow 1,000 (Teissandier et al, 2019). Transposable element expression was quantified with featuresCount from the Subread package version 2.0.1 (Liao et al, 2014) using annotation from RepeatMasker downloaded from the UCSC archive (mm10 build). Differential expression was assessed with DESeq2 version 1.30.1 (Love et al, 2014).

The analysis pipeline was wrapped in a Snakemake pipeline to automate execution (Mölder et al, 2021). All the analysis software has been containerized, and Singularity recipes are distributed together with the analysis code.

Tertiary analyses were performed in a separate environment using R version 4.1.0 (Team RC, 2020) and Bioconductor version 2.52 (Huber et al, 2015). Protein-coding genes showing significant differences between conditions were spit into up-regulated and down-regulated according to the observed $\log_2$ fold change. Gene ontology (Ashburner et al, 2000) enrichment analysis was performed for each gene set using the Bioconductor package clusterProfiler version 4.0.5 (Yu et al, 2012).

Transcripts Per Million expression of *Trim66* in the oocyte and at key stages of preimplantation development was computed from the mRNA-seq dataset GSE66582 (Wu et al, 2016).

## Quantification of GFP expression

RNA-seq libraries were realigned to a modified version of the mm10 genome (UCSC mm10). The reverse complement of eGFP protein was manually inserted at chr7:109,484,668–109485543. This modified genome was used to build the STAR index (2.7.11a). Reads were aligned on this reference with the following parameters: --runMode alignReads --outSAMtype BAM SortedByCoordinate --outTmpDir $TEMPDIR/STAR --runThreadN $SLURM_NTASKS --outBAMsortingThreadN $SLURM_NTASKS --genomeDir $GENOME --readFilesCommand zcat --outFileNamePrefix $PREFIX --limitBAMsortRAM $(($SLURM_MEM_PER_NODE * 1,000,000)) --genomeLoad NoSharedMemory --readFilesIn $M1 $M2 --bamRemoveDuplicatesType UniqueIdentical --outBAMcompression −1. Genome-aligned results were indexed with SAMtools index (1.18), and the coverage of the eGFP insert was quantified with SAMtools coverage as follows: samtools coverage --region chr7: 109484669–109485543 "${PREFIX}Aligned.sortedByCoord.out.bam" > "${PREFIX}coverage.txt." Library size was estimated using Picard (3.1.0) CollectAlignmentSummaryMetrics with the following parameters: R=$GENOME I=$BAM O=${BAM/bam/alignment_metrics.txt}, where $GENOME represents the path to the modified mm10 sequence, and $BAM represents the path to each alignment in genomic coordinates from STAR. A custom R script was developed to calculate eGFP FPKM for each sample as follows:

$FPKM_i = RC_i/1 \times 10^6/len(GFP)/1 \times 10^3$ where $RC_i$ represents the read count of each i-th sample. Statistical difference between mutant and WT samples was assessed with the Wilcoxon–Mann–Whitney rank sum test.

## ChIP-seq data analyses

Six sequencing libraries were analyzed (three biological replicates per genotype). Their quality was assessed with FastQC version 0.11.9. Sequencing adapters were removed with Trimmomatic version 0.36 [12] with the following parameters: ILLUMINACLIP:TruSeq3-SE.fa:2:0: 10 LEADING:28 MINLEN:40. Trimmed reads were aligned to the mouse genome (Gencode GRCm38 M25/mm10) using Bowtie2 version 2.5.0 (Langmead & Salzberg, 2012) with the following parameters: --local --very-sensitive --no-mixed --no-discordant --dovetail. Duplicated reads were marked and removed with Picard MarkDuplicates version 2.27.4. Reads mapping to problematic regions as defined by the Encode project (Amemiya et al, 2019) were discarded with SAMtools view version 1.16.1 (Danecek

et al, 2021) with the following parameters: -F 0x004 -q 1 -L <encode_blacklist_complement>, where <encode_blacklist_complement> represents a path to a bed file with included genomic regions. This bed file was computed from the original Encode exclusion list (https://www.encodeproject.org/annotations/ENCSR636HFF/) using bedtools complement version 2.30.0 (Quinlan & Hall, 2010). Reads with more than four mismatches and mapping quality lower than 20 were removed with BamTools version 2.5.2 (Barnett et al, 2011). Resulting alignment files were sorted with SAMtools sort and indexed with SAMtools index (version 1.16.1); bigwig files were computed with Deeptools bamCoverage tool version 3.5.1 using the RPGC normalization method and setting bin size to 1 bp. Immunoprecipitation quality was assessed on the resulting alignments with Deeptools' plotFingerprint command version 3.5.1 (Ramírez et al, 2016). Fragment size was estimated with the csaw R package version 1.32.0 (Lun & Smyth, 2016) by estimating the cross-correlation between signals on the plus and minus strands after shifting reads of a given amount of base pairs. The fragment size was determined as the shift size maximizing the cross-correlation value. Peaks were called with MACS2 callpeak version 2.2.7.1 (Zhang et al, 2008) and the following parameters: --qvalue 0.01 --mfold 10 50 --fix-bimodal --keep-dup all --extsize <estimated_fragsize> --gsize mm. Peak summits were refined with MACS2 refinepeaks version 2.2.7.1 (Zhang et al, 2008). Peaks were annotated with Homer annotatePeaks.pl version 4.11 (Heinz et al, 2010) against the Gencode vM25 annotation. Peaks were visualized with a genestack plot using Deeptools computeMatrix and plotHeatmap tools version 3.5.1.

To compare peak intensity across conditions, peak summits were retrieved from MACS2 refinepeaks output. Standardized peaks were computed with bedtools slop version 2.30.0 setting the extension size to be 200 bp on both sides, thus yielding 401-bp-long peaks. The agreement for each possible combination of samples was computed with Intervene version 0.6.5 (Khan & Mathelier, 2017). Peak profiles were plotted using the RPGC-normalized bigwig files, MACS2 peak coordinates, and IGV version 2.13.2 (Thorvaldsdóttir et al, 2013).

## Visualization of scRNA-seq via Uniform Manifold Approximation and Projection

Data from A and B were processed similar to Shami et al (2020). Briefly, processed Drop-seq human and mouse data and cell metadata were downloaded from GEO (GSE142585 and GSE112393, PubMed: 32504559 and 30146481, respectively). A one-to-one mapping between human and mouse gene IDs was generated with Ensembl's BioMart online tool (Ensembl version 104, March 2022). Orthologous genes were mapped to the first alphabetical match. Gene count matrices were converted to Seurat (version 4.1.0, R version 4.1.0, Bioconductor version 2.52) objects and normalized using the SCTransform function from Seurat. Principal component analysis was computed independently for each dataset. The two datasets were integrated using canonical correlation analysis. Principal component analysis was recomputed for the new integrated dataset and Uniform Manifold Approximation and Projection

computed using the first 12 principal components. The embeddings were finally plotted and colored using cell labels from previous analyses.

### Protein purification

The DNA sequence encoding the murine TRIM66 PHD-Bromodomain (a.a. 996–1,185) was synthesized (IDT) and subcloned into the pGEX-6P-1 vector containing an N-terminal glutathione S-transferase (GST) tag. The recombinant protein was produced in *E. coli* strain BL21(DE3) (Novagen). The cells were grown in LB medium at 37°C, and the protein expression was induced at OD600 of 0.7 with 0.2 mM IPTG (isopropyl $\beta$d-1-thiogalactopyranoside) and 100 mM $ZnSO_4$. The cells were further incubated overnight at 16°C. The cells were harvested, resuspended in lysis buffer (50 mM Tris, pH 8.0, 500 mM NaCl, 5 mM DTT) supplemented with 0.01 mg/ml DNase, 5 mM $MgCl_2$, and 1x cOmplete protease inhibitor cocktail (Roche), and lysed using Microfluidizer. After centrifugation at 4°C for 45 min at 140,000$g$, the cleared lysate was loaded onto a 5-ml Protino GST/4B column (Macherey-Nagel) and washed with 10 column volumes of lysis buffer. The GST tag was cleaved overnight on-column at 4°C using $His_6$-3C protease. The $His_6$-3C protease was then removed from the sample by immobilized metal affinity chromatography using a 1-ml Protino Ni-NTA column (Macherey-Nagel). The mTRIM66 PHD-Bromodomain protein was further purified by size-exclusion chromatography using a HiLoad 16/600 Superdex 75 pg column (Cytiva) in 20 mM Tris, pH 7.5, 500 mM NaCl, and 5 mM DTT buffer.

### Isothermal titration calorimetry

ITC titrations were performed with a MicroCal PEAQ-ITC (Malvern Panalytical GmbH) at 25°C. Murine TRIM66 PHD-Bromodomain was dialyzed overnight at 4°C against ITC buffer (20 mM Tris, pH 7.5, and 100 mM NaCl). Lyophilized peptides (H3 (1–30) unmodified, H3 (1–30) K4me3, H3 (1–30) K4me3+K18Ac, H3 (1–30) K9me3, H3 (1–30) K9me3+K18Ac, H3 (1–30) K18Ac, H3 (1–30) K23Ac, H3 (1–30) K27Ac; Peptide Specialty Laboratories GmbH, Table S3) were resuspended in the ITC buffer, and the pH was adjusted to 7.5. Solutions of 30–50 $\mu$M of mTRIM66 PHD-Bromodomain in the cell were titrated by injection of 800 $\mu$M to 2.4 mM of peptide in the syringe. The experiments were performed in triplicate. Control experiments (buffer into protein and peptide into buffer) were also performed. The fitted offset option, in PEAQ-ITC analysis software, was used to correct for the heat of dilution. The data were fitted using a single-site binding model and analyzed using MicroCal PEAQ-ITC analysis software.

### Statistical analyses

Statistical testing was performed with R version 4.1.1 (2021-08-10). The graphs were plotted with ggplot2 3.3.5. No statistical methods were used to predetermine the sample size. The statistical tests used in this study are indicated in the respective figure legends: non-significant (*n.s.*): $P > 0.05$; *$P < 0.05$; **$P < 0.01$; ***$P < 0.001$; ****$P < 0.0001$.

## Data and Code Availability

RNA-seq data have been deposited in ArrayExpress and are available under the accession E-MTAB-12471. ChIP-seq data have been deposited in ArrayExpress and are available under the accession E-MTAB-13088. This study does not report the original code. The software and algorithms supporting this study are available at the dedicated GitHub repository: https://github.com/boulardlab/trim66-testis. Fine-grained results from analysis of breeding, sperm parameters, and RNA-seq data can be accessed at: https://boulardlab.github.io/trim66-testis/.

## Supplementary Information

## Acknowledgements

We would like to thank EMBL Facilities, in particular the Laboratory Animal Resources Facility, Gene Editing & Embryology Facility, Microscopy Facility, Histology, Bioinformatics Services, and the Genomic Core Facility. We also thank Agnieszka Sadowska for assistance in the management of this project, Emerald Perlas for useful advice and assistance with histology, Monica Di Giacomo for invaluable technical advice with flow cytometry and histology of male germ cells, Mustapha Oulad-Abdelghani for the gift of the monoclonal antibody anti-TRIM66, and Ana Boskovic for comments on the study. This research was funded by a European Molecular Biology Laboratory (EMBL) program grant to M Boulard.

### Author Contributions

M Mielnicka: conceptualization, resources, data curation, formal analysis, validation, investigation, visualization, methodology, and writing—original draft, review, and editing.
F Tabaro: conceptualization, data curation, software, formal analysis, validation, investigation, visualization, methodology, and writing—original draft, review, and editing.
R Sureka: investigation, methodology, and writing—original draft, review, and editing.
B Acurzio: validation, investigation, visualization, methodology, and writing—review and editing.
R Paoletti: investigation and methodology.
F Scavizzi: formal analysis, investigation, and methodology.
M Raspa: investigation and methodology.
AH Crevenna: investigation, visualization, and methodology.
K Lapouge: formal analysis, validation, investigation, visualization, and methodology.
K Remans: formal analysis, supervision, validation, investigation, visualization, methodology, and writing—original draft.
M Boulard: conceptualization, formal analysis, supervision, funding acquisition, investigation, visualization, methodology, project administration, and writing—original draft, review, and editing.

## Conflict of Interest Statement

The authors declare that they have no conflict of interest.

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
