## [Reviewer comments · Life Science Alliance]

Trim66's paternal deficiency causes intrauterine overgrowth

Monika Mielnicka, Francesco Tabaro, Rahul Sureka, Basilia Acurzio, Renata Paoletti, Ferdinando Scavizzi, Marcello Raspa, Alvaro Crevenna, Karine Lapouge, Kim Remans, and Matthieu Boulard

DOI: <https://doi.org/10.26508/lsa.202302512>

Corresponding author(s): *Matthieu Boulard, European Molecular Biology Laboratory*

Review Timeline:	Submission Date:	2023-12-06
	Editorial Decision:	2024-01-11
	Revision Received:	2024-04-19
	Editorial Decision:	2024-04-22
	Revision Received:	2024-04-24
	Accepted:	2024-04-25

Transaction Report:

January 11, 2024

Re: Life Science Alliance manuscript #LSA-2023-02512-T

Dr. Matthieu Boulard
European Molecular Biology Laboratory
Epigenetics & Neurobiology Unit
EMBL Rome
Via Ramarini 32
Monterotondo 00015
Italy

Dear Dr. Boulard,

Thank you for submitting your manuscript entitled "Trim66's paternal deficiency causes intrauterine overgrowth" to Life Science Alliance. The manuscript was assessed by expert reviewers, whose comments are appended to this letter. We invite you to submit a revised manuscript addressing the Reviewer comments.

Thank you for this interesting contribution to Life Science Alliance. We are looking forward to receiving your revised manuscript.

Sincerely,

B. MANUSCRIPT ORGANIZATION AND FORMATTING:

Reviewer #1 (Comments to the Authors (Required)):

In their manuscript, Mielnicka et al. investigated the role of TRIM66 during male reproduction. TRIM66 is a protein that is expected to be associated with the chromatin (it contains domains that allow interaction with histone H3 and HP1) and is predominantly expressed in postmeiotic male germ cells (i.e. elongating spermatids). The authors generated two types of loss of function (KO) mouse models, Trim66Gfp/Gfp and Trim66 Phd/Phd, which respectively carry an insertion of the Gfp coding sequence at the beginning of Trim66 gene and a stop codon mutation at the end of the gene in the region encoding the PHD domain. Intriguingly, while the KO does not lead to any visible impairment of spermatogenesis, the progeny born from KO males have a significant higher weight at birth. The subsequent molecular analyses focus on one line (the GFP one). RNAseq were performed on round spermatids and on elongating spermatids. In KO round spermatids, ~250 genes are found deregulated in the KO; Several deregulated genes encode proteins involved in H3K4me3 regulation. The authors also performed H3K4me3 ChIP-Seq on spermatozoa - no significant changes were observed in KO vs CTL samples. The molecular mechanism by which Trim66 loss of function in the paternal germline impairs progeny development remains therefore to be identified.

The article is clear, well written and the figures are nicely presented. The topic, i.e. the transmission of epigenetic defects from father to offspring, is a fascinating and timely one that deserves attention. However, I do have some comments and questions that should help to clarify some key aspects of the results.

1. TRIM66 expression and interaction with H3 unmethylated at K4 (K9me3, K18ac) are convincing.
 - It would be useful to cite the article by Khetchoumian et al 2004, which describes the TRIM66 expression pattern in detail. NB. Trim66 RNA seems to be abundant in round spermatids, whereas the protein is only visible during spermatid elongation. It would be interesting to comment on this observation. Perhaps it is regulated by a post-transcriptional mechanism, like protamine genes?
 - On the contrary to Fig1e, Fig 1C shows condensed rather than elongating spermatids (since round spermatids are observed in the same tubule). Could you correct the text or show another IHC picture?
2. TRIM66 loss of function
 - Please show full length WB images to confirm that no truncated form of the protein is still produced.
 - To try to explain the observed (small) discrepancy in testis weight of the progeny sired by the line vs the phd line, did the authors test for the presence of alternative transcripts or shorter TRIM66 protein in the phd line?
 - GFP expression could also contribute to the observed differences between the 2 lines. Yet, the authors did not detect any GFP by FACS. I think it is a critical point that requires to be confirmed and discussed:
 - Did the authors include a positive control in their FACS experiment? Were RNAseq data checked for the presence of GFP transcript? Did they also try to detect GFP by immunofluorescence?
 - How do the authors explain the absence of GFP expression in Trim66gfp/gfp ? degradation of the mRNA?
 - Another potential explanation for the observed differences (that would deserve to be mentioned) is the fact that the 2 lines were produced on different backgrounds (FVBN vs C57Bl6). Were the males then backcrossed or maintained on their initial background for all the experiments?
3. Characterization of the spermiogenesis phenotype of KO lines:
 - I assume that the authors looked at sperm morphology (in particular spermhead) but it is not mentioned: Was the morphology of spermatozoa from KO lines normal?
4. Pup weight: were there differences between males vs female pup weight? Was the number of litters and litter size similar between KO and control ? Was the phenotype similar if homozygous KO males were mated to homozygous KO females?
5. RNA-seq data. More information is needed. Figure 4 legend only mentions round spermatids (RS) and not elongating spermatids (ES). The figure shows 2 downregulated genes. Which gene is downregulated along with Trim66 ? Please also add in the figure legend the number of samples analyzed by RNAseq: 6 vs 6 for round spermatids (RS). 3 vs 3 for ES?

- Figure S5 could be explained with more details. Besides, the authors could mention the purity of RS and ES samples used for RNAseq. Could the authors provide a picture of the cells that were sorted and/or a FACS graph? This is an important question, since almost no genes were found deregulated in ES, while one would expect to see a stronger effect in ES (where Trim66 is predominantly expressed) rather than in RS. The sorted ES could actually be a more heterogeneous group of cells than RS (and only the morphology of the sorted cells could give an indication). It would be useful to include PCA of all RNAseq samples.
 - GSEA (Gene Set Enrichment Analysis) could be a nice complementary analysis to perform on RS and ES RNA-seq data as it includes all genes without threshold.
 - It is rather counterintuitive to find gene deregulation in Trim66 KO round spermatids when TRIM66 protein is not visible at this stage, could the authors comment on this? Could it be that early elongating spermatids (stage IX) are sorted together with RS? Again, a better characterisation of the cells sorted by FACS and used for analysis could provide information.
6. ChIP-seq: Please provide a bit more info on ChIP-seq results, such as genome browser capture. Only few (less than 100?) peaks were obtained but ChIPseq on spermatozoa is still controversial so I will not dwell on it. It would be worth mentioning sperm purity % though.
- NB. I would not mention "minor alterations of H3K4me3 patterns on sperm retained nucleosomes." But rather no alteration... Since it was not significant.

Overall, I understand why the authors first looked at H3K4me3 in spermatozoa, but given their RNA-seq results, wouldn't it be more relevant to compare H3K4me3 profiles in KO vs. control RS/ES?

7. Mechanism of intergenerational transmission. This is the weakest part of the study, as the authors only investigated H3K4me3 as a carrier of father to offspring epigenetic transmission (and no significant differences were found).

- Other potential carriers were not investigated, in particular small non coding RNA which have been demonstrated to be essential in other studies.

- Also the authors did not seem to consider that Trim66 could be an imprinted gene, or regulate imprinted genes. Could they develop this part of the discussion?

Additional minor comments:

- In the introduction, "firstly, the silencing of transcription genome-wide that is likely important to repress harmful genomic parasites thus preventing their proliferation in the genome of the sexual population (Ernst et al, 2019; Bestor, 2003)." I don't think these 2 references suggest that the global transcriptional shutdown at the end of spermatogenesis is related to the repression of selfish/repeat genetic elements. The repression of selfish/repeat genetic elements occurs much earlier during spermatogenesis. Please check again. Most articles hypothesize that this genome-wide transcriptional silencing is a consequence of sperm nuclear compaction (which itself is required for sperm motility and possibly to protect the paternal genome from damage).

- Trim66 tag construct is presented in the methods but I did not see any reference to this line in the result section.

- In the discussion "Hence, an alternative possibility to the alteration of the modifications of histones retained in the sperm is the transmission of extracellular vesicles in the seminal fluid. ». Transmission of vesicles could indeed be a carrier of intergenerational transmission but not necessarily linked to histones.

-typos to correct:

"spermatids-specific expression".

"while Zho et al. found a specific binding to unmodified H3K4, tri-methylated K9, and acetylated K18 (Zuo et al, 2022)."

Reviewer #2 (Comments to the Authors (Required)):

Mielnicka and colleagues set out to explore the function of TRIM66 in mice using reverse genetics. The authors provided a careful characterization of Trim66 activity in reproductive tissues of the mouse and produced two independent mouse knockout models. Trim66 knockout mice had no overt developmental phenotype besides a statistically significant paternal effect on the growth of foetuses sired by knockout males. The authors attempt to explore why that is by probing the transcriptome of round and elongating spermatids, without finding strong gene differential expression. Nevertheless, genes related to the methylation of the lysine 4 of histone H3 (H3K4me) were more abundant among the 75 upregulated genes suggesting that aberrant histone methylation in sperm could be linked to the observed paternal effect. However, ChIP-sequencing of H3K4me3 could not uncover any statistically significant differences between Trim66 knockout sperm when compared to wild type.

Overall, the study is carefully conducted and with high quality. It's the first one to tackle the description of Trim66 in a developmentally relevant context in vivo, and despite the authors not finding any overt developmental defect, it is an important contribution that should be published.

I have a few comments and suggestions:

- 1) Zuo et al., 2022 (Cell Stem Cell) proposed that Trim66 knockout mouse embryonic stem cells have an expanded potential due to the role of TRIM66 in silencing the 2-cell-like program and the resulting totipotent state. Considering the authors' observations, it would be important to discuss if the paternal effect could be due to an expanded totipotent stage in pre-implantation heterozygous embryos (pointing out that the dose would matter in this case). It would be also important to highlight that despite the roles described in ESCs by Zuo et al. a potential role for TRIM66 in turning off the 2-cell program in vivo does not seem to be essential for development. Finally, it would be essential the authors describe if breeding pairs from homozygous KO females with homozygous KO males are fertile and their progeny are viable.
- 2) Given the narrow phenotypic differences between pups sired by homozygous KO fathers versus wild type ones, it is essential the authors rule out any secondary genetic effects that may affect the overall performance of genetically modified mice instead of a Trim66-specific phenotype. Please provide information on backcrosses after strain generation.
- 3) Please provide the FACS profiles/strategy for the sorting of spermatids.
- 4) As a suggestion, ChIP may not be as quantitative as a western blot. Perhaps the authors can better address the question of any H4K4me quantitative changes by probing the sperm proteome for H3K4me1/2/3.

Reviewer #3 (Comments to the Authors (Required)):

Trim66 mRNA is only detected in post-meiotic germ cells, and interacts with H3K4 without methylation. The author created two loss-of-function mouse lines of Trim66, which are viable, and fertile without any obvious phenotypes. The neonatal weights of pups from homozygous fathers are heavier than those from wild-type fathers, but the weight difference is reduced by the time of weaning. KO round spermatids have elevated H3K4 methyltransferases, but no significant changes in H3K4me3 are observed in KO sperm. Although the current study showed primarily negative results, including minor phenotype in KO animals and minimum measurable changes in KO sperm, the study provides solid knowledge of Trim members in sperm. The experimental design is logical and the experiments were well executed.

I only have some minor questions:

In the mating setup in Figure 3, all heterozygous pups are heavier than WT pups. How about the weight of the neonatal weight of KO pups? I would suggest the authors use WT males to KO female mating as the control groups, so that the pups are heterozygous same as those in the experimental group. How is the placental weight?

Some data are presented as mean +/- numbers. The author should describe the data format, like Mean +/- SEM or variance.

Is Trim66 an imprinted gene? Any expression differences of Trim66 levels comparing maternal and paternal Trim66 allele in the placenta?

The phenotype is not very strong. Whether the difference in body weight continues in adulthood in the Trim66gfp group? Do heterozygous pups from homozygous fathers develop obesity in adulthood?

I would suggest the authors tone down the title. Like "Paternal deficiency in Trim66 causes minor intrauterine overgrowth."

We thank the expert reviewers for their careful and insightful review of our manuscript. We were pleased that reviewer 1 commented: "*The article is clear, well written and the figures are nicely presented. The topic, i.e. the transmission of epigenetic defects from father to offspring, is a fascinating and timely one that deserves attention*", that reviewer 2 finds that "*the study is carefully conducted and with high quality. It's the first one to tackle the description of Trim66 in a developmentally relevant context in vivo, and despite the authors not finding any overt developmental defect, it is an important contribution that should be published*", and that reviewer 3 stated "*the study provides solid knowledge of Trim members in sperm*".

Along with the revised manuscript, we provide the source data and the code used for statistical analyses. During the revision, we uncovered an error that occurred in the table transformation for data analyses (original submission Fig 3A, right panel), whereby empty missing values (NA) were introduced in the data at weaning. We corrected this error, and we report that the only conclusion that changed in the weight at weaning of progeny of *gfp/gfp* father that is not significant (new Fig S5A). None of the other results were affected. The highly significant *p* values for the weight at birth of pups sired by *Trim66*-mutant males remained unchanged, our main conclusion holds strong. Furthermore, we measured the weight at birth of pups produced by the intercrosses of homozygous males with homozygous females and observed the predicted overweight phenotype at birth with very high significance. Hence, we independently replicated the key finding of this study.

We have addressed each of the reviewers' comments below; the original comments are in blue italic, our responses in Roman.

Reviewer #1 (Comments to the Authors (Required)):

In their manuscript, Mielnicka et al. investigated the role of TRIM66 during male reproduction. TRIM66 is a protein that is expected to be associated with the chromatin (it contains domains that allow interaction with histone H3 and HP1) and is predominantly expressed in postmeiotic male germ cells (i.e. elongating spermatids). The authors generated two types of loss of function (KO) mouse models, Trim66Gfp/Gfp and Trim66 Phd/Phd, which respectively carry an insertion of the Gfp coding sequence at the beginning of Trim66 gene and a stop codon mutation at the end of the gene in the region encoding the PHD domain. Intriguingly, while the KO does not lead to any visible impairment of spermatogenesis, the progeny born from KO males have a significant higher weight at birth. The subsequent molecular analyses focus on one line (the GFP one). RNAseq were performed on round spermatids and on elongating spermatids. In KO round spermatids, ~250 genes are found deregulated in the KO; Several deregulated genes encode proteins involved in H3K4me3 regulation. The authors also performed H3K4me3 ChIP-Seq on spermatozoa - no significant changes were observed in KO vs CTL samples. The molecular mechanism by which Trim66 loss of function in the paternal germline impairs progeny development remains therefore to be identified.

The article is clear, well written and the figures are nicely presented. The topic, i.e. the transmission of epigenetic defects from father to offspring, is a fascinating and timely one that deserves attention.

We thank the reviewer for their appreciation of our work and its broader implications.

However, I do have some comments and questions that should help to clarify some key aspects of the results.

1. TRIM66 expression and interaction with H3 unmethylated at K4 (K9me3, K18ac) are convincing.

We are pleased that the reviewer finds these results compelling.

- It would be useful to cite the article by Khetchoumian *et al* 2004, which describes the TRIM66 expression pattern in detail. NB. Trim66 RNA seems to be abundant in round spermatids, whereas the protein is only visible during spermatid elongation. It would be interesting to comment on this observation. Perhaps it is regulated by a post-transcriptional mechanism, like protamine genes?

We agree and included the citation Khetchoumian *et al.* 2004 in the revised manuscript: "In good agreement with a previous report, we detected TRIM66 protein in elongated spermatids while no TRIM66 staining was observed in round spermatids (Khetchoumian *et al.*, 2004)."

The reviewer is correct: the data presented show that *Trim66* is first transcribed in round spermatids and mRNA remains high in elongated spermatids (Figs 1B,C). By immunostaining we detected TRIM66 protein uniquely in elongated spermatids, in good agreement with Khetchoumian *et al.* 2004 (Fig 1E). It cannot be excluded that TRIM66 protein could be present below the detection threshold in round spermatids (discussed below).

- On the contrary to Fig1e, Fig 1C shows condensed rather than elongating spermatids (since round spermatids are observed in the same tubule). Could you correct the text or show another IHC picture?

We apologize for this error that we have corrected.

This comment underlines that Fig 1C and Fig 1E show the same result with different tools. Thus, we replaced the former Fig 1C by a new transcriptomic analysis of *Trim66* expression in male and female germ cells and in all key stages of preimplantation development. To our knowledge, *Trim66* expression in the embryo has not been reported before. The data confirms that *Trim66* is expressed in a male's germ cells, but not in the oocyte. In the embryo, *Trim66* was only detectable at the 2-cell stage, albeit at much lower levels than in spermatids. This result provides additional evidence that the paternal effect phenotype observed is almost certainly caused by a molecular defect occurring in the father's germ cells (discussed below).

2. TRIM66 loss of function

- Please show full length WB images to confirm that no truncated form of the protein is still produced.

The revised manuscript includes the uncropped WB images (Figure S4B,C), which show that there is no smaller form of TRIM66 when comparing wild type and homozygous samples.

- To try to explain the observed (small) discrepancy in testis weight of the progeny sired by the line vs the *phd* line, did the authors test for the presence of alternative transcripts or shorter TRIM66 protein in the *phd* line?

This comment is probably about "pups weight" rather than "testis weight" (the testis weight being normal in adult *Trim66*-KO (Figs 3G and Fig S6B)).

We analyzed the impact of both *Trim66* mutations on the splicing of *Trim66* mRNA: In the new Figure S4A we provide a sashimi plot for the *phd* mutation, and Figure 2A shows a sashimi plot for the *gfp* mutation. These analyses reveal that there is no alternative *Trim66* transcript produced in either homozygous mutants. In light of these results, the higher significance of the overgrowth phenotype for the *gfp* mutation as compared to the *phd* mutation is almost certainly explained by the difference of genetic background (FVB VS. C57Bl6, discussed further below).

- GFP expression could also contribute to the observed differences between the 2 lines. Yet, the authors did not detect any GFP by FACS. I think it is a critical point that requires to be confirmed and discussed:

- Did the authors include a positive control in their FACS experiment? Were RNAseq data checked for the presence of GFP transcript? Did they also try to detect GFP by immunofluorescence?

In the revised manuscript, we quantified *gfp* mRNA expression in round and elongated spermatids (Fig S3A). The data show that chimeric mRNAs including *gfp* are expressed in *Trim66^{gfp/gfp}* homozygous round and elongated spermatids (but not in wild type).

We have repeated the flow cytometry experiment, including a positive control (Fig S3B). As a positive control for *gfp* expression, we used the D4/XEGFP strain that carries a CMV transgene inserted on the X-chromosome, thus expressed in all male cells (DOI: 10.1038/893). As expected, testicular cells isolated from males hemizygous for D4/XEGFP emit *gfp*-fluorescence. While no fluorescence was detected in testicular cells isolated from males homozygous for *Trim66-gfp*, similarly to wild type controls. Hence, the new data confirms that the *Trim66-gfp* reporter is not translated.

We also attempted to detect GFP fluorescence on testis sections using fluorescence microscopy and did not observe any signal from homozygous *Trim66^{gfp/gfp}*. Because flow cytometry is more sensitive and quantitative than fluorescence microscopy, we included only the most authoritative evidence, namely flow cytometry.

Importantly, the lack of detectable *gfp* fluorescence (Fig S3B) rules out the hypothesis that expression of ectopic GFP protein could explain the slightly different severity of the overgrowth phenotype between *gfp* and *phd* mutations.

- How do the authors explain the absence of GFP expression in *Trim66gfp/gfp* ? degradation of the mRNA?

The presence of *gfp* mRNAs in RS *Trim66^{gfp/gfp}* (Fig S3A), excludes the hypothesis of a degradation of *gfp* mRNA. Thus, the absence of GFP signal in the homozygous *gfp/gfp* testicular cells is likely due to a failure of translation of the *gfp* chimeric transcript. One possibility is that translation in RS and ES requires a specific signal in the 3'UTRs, as previously reported for Protamine 1 (Braun RE, Genes Dev, 1989). The chimeric *gfp* mRNA lacks the natural 3'UTR of *Trim66* (see Fig 2A), which could in principle explain the lack of translation in spermatids. We mention this hypothesis in the revised manuscript:

"Hence, the *egfp* mRNA transcribed from the *Trim66^{gfp}* allele appeared not to be translated *in vivo*, possibly because of the lack of proper 3' UTR in the *egfp* chimeric transcript (Fig 2A) (Braun et al, 1989) "

- Another potential explanation for the observed differences (that would deserve to be mentioned) is the fact that the 2 lines were produced on different backgrounds (FVBN vs C57Bl6). Were the males then backcrossed or maintained on their initial background for all the experiments?

This is the most likely explanation in light of the new evidence provided. The new results discussed above show that both *phd* and *gfp* alleles do not produce the TRIM66 protein nor a truncated form; therefore, both mutations are considered as null mutations (complete absence of TRIM66 protein in homozygous animals). As a result, it is very likely that the small difference in severity of the overgrowth phenotype at birth is caused by the different genetic backgrounds, as stated in the revised manuscript:

"The variable severity of the phenotype could be caused by the different genetic backgrounds of the two loss-of-function strains (FVB VS. C57Bl6j)."

In the methods section, it is specified that both *Trim66* mutations were created by microinjection of Cas9, sgRNA and dsDNA donor template in the mouse zygotes. The genetic background of the zygotes used was FBV or C57Bl6j as specified in the methods section. We now made explicit in the method section that "The *Trim66^{gfp}* strain was maintained on a FVB/NCrl genetic background, the *Trim66^{phd}* strain was maintained on a C57Bl/6j background".

3. Characterization of the spermiogenesis phenotype of KO lines:

- I assume that the authors looked at sperm morphology (in particular spermhead) but it is not mentioned: Was the morphology of spermatozoa from KO lines normal?

The morphology of spermatozoa produced by *Trim66^{gfp/gfp}* animals was indistinguishable from that of wild type littermate. Micrographs are provided in Fig S6A.

4. Pup weight: were there differences between males vs female pup weight? Was the number of litters and litter size similar between KO and control ? Was the phenotype similar if homozygous KO males were mated to homozygous KO females?

The sex of the F1 pups were not recorded. There was no significant difference in litter size when homozygous males or females were crossed with a wild type animal of the opposite sex (Figs 3C,D,F).

We performed the experiment suggested and intercrossed homozygous KO males with homozygous KO females: the progeny was born overweight ($p=4.94 \times 10^{-6}$, two-sided *t*-test, Fig S5C). This result was expected given the paternal effect phenotype reported. However, we found that the litter size produced by this cross was significantly reduced ($p=0.00678$, two-sided *t*-test, Fig S5D).

5. RNA-seq data. More information is needed. Figure 4 legend only mentions round spermatids (RS) and not elongating spermatids (ES). The figure shows 2 downregulated genes. Which gene is downregulated along with Trim66 ? Please also add in the figure legend the number of samples analyzed by RNAseq: 6 vs 6 for round spermatids (RS). 3 vs 3 for ES?

We apologize for the incomplete legends of Figure 4, which we have now corrected. We provide in Table S1 the list of differentially expressed genes. As now reported in Table S1, the downregulated genes along with *Trim66* in ES is *Aqp2* (note the very low expression level).

We included in the revised figure legend the number of biological replicates analyzed.

- Figure S5 could be explained with more details. Besides, the authors could mention the purity of RS and ES samples used for RNAseq. Could the authors provide a picture of the cells that were sorted and/or a FACS graph? This is an important question, since almost no genes were found deregulated in ES, while one would expect to see a stronger effect in ES (where Trim66 is predominantly expressed) rather than in RS. The sorted ES could actually be a more heterogeneous group of cells than RS (and only the morphology of the sorted cells could give an indication). It would be useful to include PCA of all RNAseq samples.

We agree that controlling for the purity of the sorting of ES and RS is paramount. We now provide four lines of evidence that demonstrates the identity of the sorted cell populations and rule out the possibility of a cross contamination:

Firstly, we include in the revised manuscript the flow cytometry data of the sorting of RS and ES (Figs S7 A-L).

Secondly, we provide in Fig S7M representative micrographs of the sorted RS and ES.

Thirdly, we included in Figure S7N a PCA showing that RS samples and ES samples separate on PCA (84.3%), evidencing that the two sorted cell population have distinct transcriptomes. This was a very useful suggestion.

Fourthly, the heat map in Figure S7O shows the expression of marker genes specific of RS and ES that were previously identified in a single cell RNA-seq experiment (Green *et al.* dev Cell 2018). This result confirms the cellular identity of the sorted cell population. The clustering of the sample excludes cross contamination between these two cell types (Fig S7O).

- GSEA (Gene Set Enrichment Analysis) could be a nice complementary analysis to perform on RS and ES RNA-seq data as it includes all genes without threshold.

The results of the GSEA are shown below. The fold change of genes contributing the most to the enrichment are very small, and therefore could reflect technical variation. We prefer to be conservative in our data analysis and not include this GSEA analysis in the manuscript.

- It is rather counterintuitive to find gene deregulation in Trim66 KO round spermatids when TRIM66 protein is not visible at this stage, could the authors comment on this?

The absence of TRIM66 protein signal assessed by immunostaining in round spermatids can have three possible causes:

1. *Trim66* mRNA is not translated in round spermatids.
2. TRIM66 protein is translated in round spermatids below the detection threshold.
3. TRIM66 is translated in round spermatids, but the TRIM66 epitope is inaccessible to antibody detection in round spermatid chromatin.

None of these possibilities can be excluded based on the available evidence.

Could it be that early elongating spermatids (stage IX) are sorted together with RS? Again, a better characterisation of the cells sorted by FACS and used for analysis could provide information.

Fig S7O shows that markers of ES are not present in sorted RS and vice versa, therefore ruling out the possibility of cross contamination between ES and RS.

We thank the reviewer for their suggestion of a better characterization of the purity and identity of the sorted RS and ES. The new results shown in Figs S7 A-N provide compelling evidence for the purity of the sort.

6. ChIP-seq: Please provide a bit more info on ChIP-seq results, such as genome browser capture. Only few (less than 100?) peaks were obtained but ChIPseq on spermatozoa is still controversial so I will not dwell on it. It would be worth mentioning sperm purity % though.

Figures 4D-F provide a comprehensive analysis of the ChIP-seq of H3K4me3 in sperm. For each of the three biological replicates, we provide the gene stack plot, the metaprofile and the number of peaks. Statistical comparison between wild type and homozygous mutant shows no difference.

We share the reviewer's views on the limitation of the ChIP-seq technique when applied to sperm and underlined this point in the revised text: "This result does not completely exclude the possibility of H3K4me3 alteration in the sperm because antibody-based approaches for epigenomic profiling of the densely condensed sperm genome are notorious to be imprecise (Yin et al, 2023)."

NB. I would not mention "minor alterations of H3K4me3 patterns on sperm retained nucleosomes." But rather no alteration.... Since it was not significant.

We agree and changed the text as followed: "Profiling of H3K4me3 patterns in the sperm produced by *Trim66*-null mutant showed minor alterations below statistical significance."

Overall, I understand why the authors first looked at H3K4me3 in spermatozoa, but given their RNA-seq results, wouldn't it be more relevant to compare H3K4me3 profiles in KO vs. control RS/ES?

The purpose of the ChIP-seq experiment in spermatozoa is to assess whether H3K4me3 patterns could be altered as a result of the upregulation of H3K4-specific HMTs in *Trim66*-deficient RS. We specifically analyzed spermatozoa because mature gamete have the potential to carry epigenetic information to the fertilized egg, and thus could explain the paternal effect phenotype.

We agree that it would be valuable to profile H3K4me3 in round spermatids and elongated spermatids as well. However, this experiment would not permit the identification of the inherited carrier that causes the overgrowth phenotype.

7. Mechanism of intergenerational transmission. This is the weakest part of the study, as the authors only investigated H3K4me3 as a carrier of father to offspring epigenetic transmission (and no significant differences were found).

- Other potential carriers were not investigated, in particular small non coding RNA which have been demonstrated to be essential in other studies.

Uncovering the molecular carrier responsible for the paternal effect overgrowth at birth is hugely challenging, considering that there is a plethora of possibilities and spermatozoa's chromatin remains scarcely characterized. We agree that small non coding RNAs represent a plausible alternative hypothesis and included this hypothesis in the discussion. Signalling molecules carried in the seminal fluid could in principle also be involved.

In this study, we tested the most obvious candidate, namely H3K4me3 in sperm, based on our transcriptomic results. Our native ChIP-seq experiment found no statistical support for this possibility. Future studies should address alternative hypotheses.

- Also the authors did not seem to consider that Trim66 could be an imprinted gene, or regulate imprinted genes. Could they develop this part of the discussion?

We thank the reviewer for this excellent suggestion. We agree that the observed paternal-effect phenotype could in principle be caused by the loss-of-function of *Trim66* in the embryo, at the necessary condition that *Trim66* were expressed in the embryo only from the paternally inherited allele (maternally imprinted).

We addressed the possibility of *Trim66* being imprinted in the preimplantation embryo by analyzing *Trim66* expression at all key cleavage stages (Fig 1C). The near absence of *Trim66* at the 4-cell stage and beyond exclude this possibility (non-canonical imprinted genes being expressed after the 4-cell stage). Genomic imprinting in the postimplantation embryo has been extensively studied and a recent comprehensive and remarkably rigorous survey for mono-allelically expressed genes in F1 hybrid mouse across 4 independent datasets has not identified *Trim66* as mono-allelically expressed (<https://doi.org/10.7554/eLife.83364>).

We have discussed this important point in the discussion: "Paternal effect phenotypes in mammals are rare and can have several possible causes (Rando, 2012; Lismer & Kimmins, 2023). For example, the loss-of-function of an imprinted gene normally expressed from the paternally inherited allele (Li et al, 1999). Recent transcriptome-wide assessments of imprinted gene expression using F1 hybrid mouse embryos, including in extraembryonic lineages, have not identified *Trim66* as mono-allelically expressed (Andergassen et al, 2021; Edwards et al, 2023). It is therefore very unlikely that *Trim66* could be imprinted itself in the embryo. The paternally imprinted gene *H19* was previously shown to act on embryonic growth but failure to methylate *H19* imprinting control region in male germ cells was shown to cause growth retardation; thus this possibility can be ruled out (Edwards et al, 2019)."

Additional minor comments:

- In the introduction, "firstly, the silencing of transcription genome-wide that is likely important to repress harmful genomic parasites thus preventing their proliferation in the genome of the sexual population (Ernst et al, 2019; Bestor, 2003)."

I don't think these 2 references suggest that the global transcriptional shutdown at the end of spermatogenesis is related to the repression of selfish/repeat genetic elements. The repression of selfish/repeat genetic elements occurs much earlier during spermatogenesis. Please check again. Most articles hypothesize that this genome-wide transcriptional silencing is a consequence of sperm nuclear compaction (which itself is required for sperm motility and possibly to protect the paternal genome from damage).

Thank you for the suggestion, we agree that the role of protamines in repressing retrotransposons remains hypothetical. Hence, we removed this sentence in the revised manuscript.

- Trim66 tag construct is presented in the methods but I did not see any reference to this line in the result section.

As explained above, the old Figure 1C - the only result produced with the *Trim66*-tag allele - was removed from the revised manuscript.

- In the discussion "Hence, an alternative possibility to the alteration of the modifications of histones retained in the sperm is the transmission of extracellular vesicles in the seminal fluid. ». Transmission of vesicles could indeed be a carrier of intergenerational transmission but not necessarily linked to histones.

We agree and make it explicit: "Hence, an alternative possibility to histone modifications for the genetic-independent paternal influence on embryonic growth could be the transmission of extracellular vesicles in the seminal fluid."

-typos to correct:

"spermatids-specific expression".

"while Zho et al. found a specific binding to unmodified H3K4, tri-methylated K9, and acetylated K18 (Zuo et al, 2022)."

We have corrected the language issues mentioned, for which we apologize.

Reviewer #2 (Comments to the Authors (Required)):

Mielnicka and colleagues set out to explore the function of TRIM66 in mice using reverse genetics. The authors provided a careful characterization of Trim66 activity in reproductive tissues of the mouse and produced two independent mouse knockout models. Trim66 knockout mice had no overt developmental phenotype besides a statistically significant paternal effect on the growth of fetuses sired by knockout males. The authors attempt to explore why that is by probing the transcriptome of round and elongating spermatids, without finding strong gene differential expression. Nevertheless, genes related to the methylation of the lysine 4 of histone H3 (H3K4me) were more abundant among the 75 upregulated genes suggesting that aberrant histone methylation in sperm could be linked to the observed paternal effect. However, ChIP-sequencing of H3K4me3 could not uncover any statistically significant differences between Trim66 knockout sperm when compared to wild type.

Overall, the study is carefully conducted and with high quality. It's the first one to tackle the description of Trim66 in a developmentally relevant context in vivo, and despite the authors not finding any overt developmental defect, it is an important contribution that should be published.

We thank the reviewer for their clear understanding of our work and their appreciation of the biological importance of the findings.

I have a few comments and suggestions:

1) Zuo et al., 2022 (Cell Stem Cell) proposed that Trim66 knockout mouse embryonic stem cells have an expanded potential due to the role of TRIM66 in silencing the 2-cell-like program and the resulting totipotent state. Considering the authors' observations, it would be important to discuss if the paternal effect could be due to an expanded totipotent stage in pre-implantation heterozygous embryos (pointing out that the dose would matter in this

case). It would be also important to highlight that despite the roles described in ESCs by Zuo et al. a potential role for TRIM66 in turning off the 2-cell program in vivo does not seem to be essential for development. Finally, it would be essential the authors describe if breeding pairs from homozygous KO females with homozygous KO males are fertile and their progeny are viable.

Thank you for raising this important point. We included new results that show that homozygous male *Trim66*-KO are fertile when crossed with homozygous female *Trim66*-KO (Fig S5C,D). However, the litter size was significantly reduced when *Trim66*-KO were intercrossed, thus providing indirect in vivo support to Zuo et al. (10.1016/j.stem.2022.05.004) proposed function for *Trim66* in repressing the 2C-program. The mouse models created in this study will be useful for future investigations of *Trim66*'s function in the embryo.

As predicted by the results shown in Fig 3 A,E, the pups produced by the intercrossing of homozygous *Trim66*-gfp were significantly overweight at birth in comparison with wild type controls (Fig S5C). This result provides an independent confirmation of the key result, which is that male lacking *Trim66* sire overweight progeny at birth.

It is noteworthy that Zuo and colleagues' data were generated using *Trim66*-KO mouse embryonic stem cells (ESCs). Importantly, no loss-of-function genetic in the mouse was used in their study. Zuo and colleagues did not provide evidence of *Trim66* expression in the embryo nor in their ESC model. To assess the consequence of their mutation on *Trim66* mRNA level, we reanalyzed their data: *Trim66* mRNA appeared in to be very low in wild type ES cells (mean wild type: 0.73 TPM, mean *Trim66*-KO: 0.42 TPM; GSE169450). Hence, it is unclear whether this ESC model is relevant to studying *Trim66*'s function.

2) Given the narrow phenotypic differences between pups sired by homozygous KO fathers versus wild type ones, it is essential the authors rule out any secondary genetic effects that may affect the overall performance of genetically modified mice instead of a Trim66-specific phenotype. Please provide information on backcrosses after strain generation.

We clarified this important point in our response to reviewer 1 and in the revised manuscript.

As specified in the methods, the mutations were created using zygotes of pure genetic background (e.g. FVB for *gfp* and C57Bl6j for *phd*). It is also specified that the strains were maintained in their original genetic backgrounds. In the revised version, we added: "All strains were maintained by breeding heterozygote animals with wild type of the same genetic background. The *Trim66^{gfp}* strain was maintained on a FVB/NCrl genetic background, the *Trim66^{phd}* strain was maintained on a C57Bl/6j background".

This breeding strategy prevents genetic drift, and the wild type colonies are regularly refreshed from the Charles River's stocks. Thus, all the experiments were performed with virtually pure and invariable genetic backgrounds.

3) Please provide the FACS profiles/strategy for the sorting of spermatids.

The FACS profiles of the sorting of round and elongated spermatids are provided in Figures S7 A-L of the revised manuscript.

4) As a suggestion, ChIP may not be as quantitative as a western blot. Perhaps the authors can better address the question of any H4K4me quantitative changes by probing the sperm proteome for H3K4me1/2/3.

The native ChIP-seq experiment concluded that H3K4me3 patterns in sperm produced by homozygous *Trim66*-null males are not significantly different from those of wild type controls. Because western-blot assess only bulk levels, it is very unlikely that this method could detect small changes that were not significant with the native ChIP-seq profiling.

Reviewer #3 (Comments to the Authors (Required)):

Trim66 mRNA is only detected in post-meiotic germ cells, and interacts with H3K4 without methylation. The author created two loss-of-function mouse lines of Trim66, which are viable, and fertile without any obvious phenotypes. The neonatal weights of pups from homozygous fathers are heavier than those from wild-type fathers, but the weight difference is reduced by the time of weaning. KO round spermatids have elevated H3K4 methyltransferases, but no significant changes in H3K4me3 are observed in KO sperm. Although the current study showed primarily negative results, including minor phenotype in KO animals and minimum measurable changes in KO sperm, the study provides solid knowledge of Trim members in sperm. The experimental design is logical and the experiments were well executed.

We thank the reviewer for their clear understanding of our work and their appreciation of its significance.

I only have some minor questions:

In the mating setup in Figure 3, all heterozygous pups are heavier than WT pups. How about the weight of the neonatal weight of KO pups? I would suggest the authors use WT males to KO female mating as the control groups, so that the pups are heterozygous same as those in the experimental group. How is the placental weight?

We included new data about the weight of homozygous pups at birth: Fig S5C shows that pups produced by the cross of KO-homozygous males with KO-homozygous females are

significantly overweight in comparison with wild type controls ($p=4.94 \times 10^{-6}$, two-sided *t*-test). This new result is in agreement with the key finding: Males lacking functional Trim66 sire pups that are overweight at birth.

The placental weight was not recorded. This was a specially interesting comment, as the placenta plays a key role in regulating embryonic growth. The placenta should be considered in follow-up studies on the developmental origin of the paternally inherited overweight phenotype.

Some data are presented as mean +- numbers. The author should describe the data format, like Mean +- SEM or variance.

The revised figure legends include a comprehensive description of the data format. We apologize for having omitted some data format description in the original submission.

Is Trim66 an imprinted gene? Any expression differences of Trim66 levels comparing maternal and paternal Trim66 allele in the placenta?

We thank the reviewer for raising this important point. All the available evidence suggests that this hypothesis is extremely unlikely. Firstly, none of the authoritative allelic-specific transcriptomic studies have not identified Trim66 as an imprinted gene (10.7554/elife.83364, 10.1038/s41588-018-0232-7). Secondly, a genetic study of maternal imprints (that causes mono-allelic expression from the paternal allele) has not found a Dnmt3l-dependent differentially methylated region near the Trim66 gene (10.1016/j.molcel.2012.07.010). Thus, previous efforts to uncover new imprinted genes provide no support to the hypothesis of an imprinted expression of Trim66. The search for new imprinted genes in the mouse is widely considered to be saturated by the imprinted field.

We discussed this possibility in the revised manuscript: "A paternal effect phenotype can in principle be caused by the loss-of-function of an imprinted genes normally expressed from the paternally inherited allele. It is very unlikely that Trim66 could be imprinted itself because recent assessments of imprinted gene expression in the mouse embryo, including in extraembryonic lineages, found no evidence to support this possibility (Andergassen et al. 2021; Edwards et al. 2023)".

The phenotype is not very strong. Whether the difference in body weight continues in adulthood in the Trim66gfp group? Do heterozygous pups from homozygous fathers develop obesity in adulthood?

We respectfully disagree with this assessment because the severity of the overgrowth phenotype is comparable to that previously reported. For example, the disruption of the imprinted gene *H19* in females results in an abnormal weight increase in the progeny of 8% (Ripoche et al. *Genes and Development*, 2007, doi: 10.1101/gad.11.12.1596). We found that pups sired by *Trim66^{gfp}* homozygous males were on average 6.9% heavier than the wild type controls, hence the severity of the overgrowth phenotype is comparable (Figure 3A).

The weight measurements were not prolonged beyond weaning due to the high cost associated with a longitudinal experiment involving several hundreds of adult mice. We have not observed signs of obesity neither for homozygous nor for heterozygote *Trim66^{gfp}* animals fed with a normal diet. Follow-up studies with different diet compositions should clarify this point.

I would suggest the authors tone down the title. Like "Paternal deficiency in Trim66 causes minor intrauterine overgrowth."

As discussed above, the severity of the intrauterine overgrowth we measured in progeny of *Trim66* fathers is of similar magnitude to that previously reported for the imprinted lncRNA *H19*, a well-known regulator of embryonic growth involved in some cases of the overgrowth disorder known as Beckwith–Wiedemann syndrome (Ripoche et al. *Genes and Development*, 2007, doi: 10.1101/gad.11.12.1596). Therefore, we prefer to keep the title neutral without qualifying the phenotype ourselves.

April 22, 2024

RE: Life Science Alliance Manuscript #LSA-2023-02512-TR

Dr. Matthieu Boulard
European Molecular Biology Laboratory
Epigenetics & Neurobiology Unit
EMBL Rome
Via Ramarini 32
Monterotondo 00015
Italy

Dear Dr. Boulard,

Thank you for submitting your revised manuscript entitled "Trim66's paternal deficiency causes intrauterine overgrowth". We would be happy to publish your paper in Life Science Alliance pending final revisions necessary to meet our formatting guidelines.

- please be sure that the authorship listing and order is correct
- please add call-outs for Figures 4F; S1A; S2A-B; S4C and S7A-O to your main manuscript text

A. FINAL FILES:

B. MANUSCRIPT ORGANIZATION AND FORMATTING:

Sincerely,

April 25, 2024

RE: Life Science Alliance Manuscript #LSA-2023-02512-TRR

Dr. Matthieu Boulard
European Molecular Biology Laboratory
Epigenetics & Neurobiology Unit
EMBL Rome
Via Ramarini 32
Monterotondo 00015
Italy

Dear Dr. Boulard,

Thank you for submitting your Research Article entitled "Trim66's paternal deficiency causes intrauterine overgrowth". It is a pleasure to let you know that your manuscript is now accepted for publication in Life Science Alliance. Congratulations on this interesting work.

DISTRIBUTION OF MATERIALS:

Again, congratulations on a very nice paper. I hope you found the review process to be constructive and are pleased with how the manuscript was handled editorially. We look forward to future exciting submissions from your lab.

Sincerely,
